# N-terminus of *Drosophila melanogaster* MSL1 is critical for dosage compensation

Valentin Babosha[1,2], Natalia Klimenko[2], Anastasia Revel-Muroz[2], Evgeniya Tikhonova[1,2], Pavel Georgiev[1]*, Oksana Maksimenko[2]*

[1]Department of the Control of Genetic Processes, Institute of Gene Biology, Russian Academy of Sciences, Moscow, Russian Federation; [2]Center for Precision Genome Editing and Genetic Technologies for Biomedicine, Institute of Gene Biology, Russian Academy of Sciences, Moscow, Russian Federation

## eLife assessment

This is a potentially **valuable** contribution, reporting a deletion analysis of the MSL1 gene to assess how different parts of the protein product interact with the MSL2 protein and roX RNA to affect the association of the MSL complex with the male X chromosome of *Drosophila*. However, the framework that the MSL complex mediates dosage compensation is outdated and has flaws, and the evidence is currently considered **inadequate** to support the claims. Because there are many ways to alter viability, sex-specific viability is insufficient to make claims regarding dosage compensation.

*For correspondence:
georgiev_p@mail.ru (PG);
maksog@mail.ru (OM)

Competing interest: The authors declare that no competing interests exist.

**Abstract** The male-specific lethal complex (MSL), which consists of five proteins and two non-coding roX RNAs, is involved in the transcriptional enhancement of X-linked genes to compensate for the sex chromosome monosomy in *Drosophila* XY males compared with XX females. The MSL1 and MSL2 proteins form the heterotetrameric core of the MSL complex and are critical for the specific recruitment of the complex to the high-affinity 'entry' sites (HAS) on the X chromosome. In this study, we demonstrated that the N-terminal region of MSL1 is critical for stability and functions of MSL1. Amino acid deletions and substitutions in the N-terminal region of MSL1 strongly affect both the interaction with roX2 RNA and the MSL complex binding to HAS on the X chromosome. In particular, substitution of the conserved N-terminal amino-acids 3–7 in MSL1 (MSL1[GS]) affects male viability similar to the inactivation of genes encoding roX RNAs. In addition, MSL1[GS] binds to promoters such as MSL1[WT] but does not co-bind with MSL2 and MSL3 to X chromosomal HAS. However, overexpression of MSL2 partially restores the dosage compensation. Thus, the interaction of MSL1 with roX RNA is critical for the efficient assembly of the MSL complex on HAS of the male X chromosome.

## Introduction

At present, it is unknown how transcription-regulating complexes bind only to certain chromatin regions that do not have a pronounced sequence specificity relative to other genome sites. Dosage compensation in *Drosophila* is a striking example of the specific recruitment of transcription complexes (***Lucchesi, 2018***). Dosage compensation equalizes the expression of different doses of X chromosomes in females (X/X) and males (X/Y), which is achieved by doubling the expression of genes on one X chromosome in males to match the level of gene expression on two chromosomes in females (***Ferrari et al., 2014***; ***Kuroda et al., 2016***; ***Lucchesi, 2018***; ***Samata and Akhtar, 2018***). Currently, there is no consensus on the mechanisms that determine dosage compensation (***Birchler and Veitia, 2021***; ***Vensko and Stone, 2015***). It is most likely that dosage compensation in males is the result

of the simultaneous effect of activation of gene transcription on the X chromosome and the inverse dosage effect, which consists of a decrease in the expression of autosomal genes (*Birchler, 2016*; *Ferrari et al., 2014*; *Kuroda et al., 2016*; *Lucchesi, 2018*; *Samata and Akhtar, 2018*; *Zhang et al., 2021*).

A multisubunit complex that binds only to the X chromosome of males was discovered (*Ferrari et al., 2014*; *Kuroda et al., 2016*; *Lucchesi, 2018*; *Samata and Akhtar, 2018*), and it was proposed that this complex is involved in equalizing the expression of genes on the X chromosomes of males and females. The complex, named male-specific lethal, consists of five proteins, male-specific lethal (MSL)1, MSL2, MSL3, males-absent on the first protein (MOF), and maleless (MLE). Two non-coding RNAs are included, namely roX1 (3.7 kb) and roX2 (0.6 kb), which perform partially redundant functions (*Kim et al., 2018*; *Kuroda et al., 2016*; *Meller and Rattner, 2002*; *Samata and Akhtar, 2018*). The proteins MSL1, MSL3, MOF, and MLE are also expressed in females. These proteins, and are involved in the regulation of gene expression processes that are unrelated to X chromosome dosage compensation (*Samata and Akhtar, 2018*). The MSL2 protein is male-specific, and the expression of the *msl-2* gene is controlled by the sex-lethal (SXL) protein, which binds to *msl-2* mRNA in females and blocks protein translation (*Beckmann et al., 2005*). Therefore, the MSL2 protein is thought to play a key role in the specific binding of the MSL complex to the X chromosome in males.

The N-terminus of MSL2 contains a really interesting new gene (RING) domain, which functions as a ubiquitin E3 ligase (*Villa et al., 2012*; *Wu et al., 2011*). MSL2 has been shown to induce the ubiquitination of itself as well as other subunits of the MSL complex. The MSL1 protein functions as a scaffold for the assembly of the MSL complex (*Gorman et al., 1993*; *Kadlec et al., 2011*; *Palmer et al., 1993*; *Scott et al., 2000*), as the N-terminal coiled-coil domain (*Hallacli et al., 2012*) of MSL1 facilitates homodimerization and interacts with the N-terminal RING domain of MSL2. The PEHE domain located in the C-terminal region of MSL1 is responsible for interactions with MSL3 and MOF (*Kadlec et al., 2011*; *Scott et al., 2000*). MSL1 and MSL2 form the structural core of the complex, whereas MLE—an ATP-dependent RNA/DNA helicase belonging to the DEAD box family—interacts with the roX1 and roX2 RNAs to induce their unwinding (*Ilik et al., 2017*; *Maenner et al., 2013*; *Meller et al., 2000*). This interaction allows both RNAs to bind MSL2 and possibly MSL1 (*Müller et al., 2020*), combining MLE, MSL2, and other MSL proteins into a single complex.

In addition to the MSL complex, lysine acetyltransferase MOF forms the catalytic core of the nonspecific lethal complex (NSL), which is recruited to promoters located on all chromosomes of both sexes and is responsible for strong transcriptional activation (*Sheikh et al., 2019*). Unlike the MSL complex, in which MOF moderately stimulates transcription during RNA polymerase II elongation (*Ferrari et al., 2014*; *Kuroda et al., 2016*), the NSL complex strongly increases the transcription at the initiation step (*Sheikh et al., 2019*). In males, formation of the MSL complex titrates MOF from the NSL complex, resulting in a decrease in its activity at all promoters (*Prestel et al., 2010*; *Sun et al., 2013*). This may be one of the reasons for the inverse dose effect, which consists of a decrease in the expression of autosomal genes (*Birchler and Veitia, 2021*).

An incomplete MSL1-MSL2 subcomplex can bind to a reproducible set of several hundreds of sites on the X chromosome, referred to as the primary chromatin entry sites (CESs) (*Alekseyenko et al., 2008*) or HAS (*Straub et al., 2008*). A GA-rich motif within the HAS/CES, which serves as an MSL recognition element (MRE), is important for the MSL complex targeting (*Alekseyenko et al., 2008*; *Straub et al., 2008*). Structural analysis and an in vitro genome-wide DNA binding assay showed that MSL2, through its CXC and proline/basic-residue-rich domains, specifically recognizes and binds to a subclass of HAS-named pioneering sites on the X (PionX) (*Fauth et al., 2010*; *Villa et al., 2016*; *Zheng et al., 2014*).

The zinc-finger protein chromatin-linked adaptor for MSL proteins (CLAMP) was found to be significant for the specific recruitment of the MSL complex to the male X chromosome (*Eggers et al., 2023*; *Larschan et al., 2012*; *Soruco et al., 2013*). CLAMP is an essential protein in *Drosophila* that binds thousands of GA-rich sequences genome-wide in both sexes (*Kuzu et al., 2016*; *Soruco et al., 2013*; *Urban et al., 2017*). In particular, CLAMP binds to GA repeats in HAS and is involved in the organization of open chromatin in these genome regions (*Albig et al., 2019*; *Urban et al., 2017*). CLAMP has an N-terminal unstructured dimerization domain (*Tikhonova et al., 2022a*) and promotes long-range chromatin interactions that mediate the dosage compensation of the male X chromosome (*Jordan and Larschan, 2021*). CLAMP also directly interacts with the MSL2 (*Albig et al., 2019*; *Eggers et al.,*

*2023*; *Tikhonova et al., 2022b*; *Tikhonova et al., 2019*) and MLE (*Quinn et al., 2014*; *Tikhonova et al., 2024*) proteins.

The CLAMP interaction and CXC domains of MSL2 have partially redundant functions for the recruitment of the MSL complex to the X chromosome (*Tikhonova et al., 2019*). A previous study (*Li et al., 2005*) showed that the N-terminal 265 amino acids region of MSL1 could bind to approximately 30 sites on polytene X chromosomes in females that exogenously express MSL2, and the first 26 aa are essential for such binding. The authors suggested that the N-terminal sequences of MSL1 can bind to DNA and facilitate the specific recruitment of MSL to the male X chromosome. However, further studies examining the mechanisms of the MSL complex recruitment to the male X chromosome have concentrated on MSL2 as this is the only dosage compensation protein that is exclusively expressed in males [see review *Kuroda et al., 2016*; *Lucchesi, 2018*; *Samata and Akhtar, 2018*]. Although MSL1 represents a core protein of the MSL complex (*Gorman et al., 1993*; *Palmer et al., 1993*; *Scott et al., 2000*), MSL1 can also independently interact with gene promoters (*Chlamydas et al., 2016*, p. 2; *Straub et al., 2013*).

Here, we tested the functional role of different regions in the N-terminal 1–85 aa region of MSL1. Deletion of the 1–15 aa results in the instability of MSL1. Unexpectedly, we found that the simultaneous replacement from 3 to 7 aa from the N-terminus or deletion from 41 to 65 aa of MSL1 strongly influences the binding of the MSL complex on the male X chromosome. These mutations also affect the interaction of MSL1 with roX2 RNA. Thus, the N-terminal region of MSL1 performs key functions in the recruitment of the MSL complex to the X chromosome.

## Results

### The N-terminal 85 amino acids of MSL1 are critical for the activity of the MSL complex

The principal goal of the work is to study the functional role of the N-terminal 85 aa MSL1 followed by the coiled-coil domain involved in homodimerization and interaction with MSL2. With the exception of the coiled-coil domain, the first 15 amino acids represent the most conserved region within the N-terminus of MSL1 among Drosophilidae (*Figure 1A*). However, among different families of Diptera (*Figure 1—figure supplement 1*), a high level of homology was only observed for the coiled-coil region, which is required for interactions with MSL2 (*Hallacli et al., 2012*). The N-terminal amino acids are also conserved among representatives within each of the Diptera families (*Figure 1—figure supplement 1*), but not across families. For example, the MSL1 of *Anopheles* mosquitoes, which are closely related to *Drosophila melanogaster*, features a completely different N-terminal domain.

To evaluate the functional effects of N-terminal domain deletions in the MSL1 protein in vivo, we created transgenic flies expressing wild-type (MSL1$^{WT}$) and mutant variants of MSL1 (MSL1$^{\Delta1-85}$, MSL1$^{\Delta1-15}$, MSL1$^{\Delta8-20}$, MSL1$^{\Delta24-39}$, MSL1$^{\Delta41-85}$, MSL1$^{\Delta41-65}$, and MSL1$^{\Delta66-85}$) (*Figure 1A*). The MSL1 variants were expressed under the control of a strong *Ubi-p63E* (Ubi) promoter (*Figure 1—figure supplement 2*). To avoid the influence of position effects on the expression of the MSL1 variants, all transgenes were inserted into the same genomic location (86F8) on the third chromosome using an φC31-based integration system (*Bischof et al., 2007*).

To determine the role played by the N-terminal sequence during the specific recruitment of the MSL complex in males, we crossed obtained transgenes into the null *msl-1⁻* background (the *msl-1⁻* background corresponds to the *msl-1^{γ269}/msl-1^{L60}* heterozygote) (*Palmer et al., 1993*). As shown previously, the main band detected in immunoblot analysis corresponds to a 170-kD protein that is considerably larger than the predicted 1039 aa protein (*Palmer et al., 1994*). Anomalous electrophoretic migration was explained by specific clusters of proline, acidic, or basic residues as well as potential post-translational modifications in particular ubiquitination. MSL1 is expressed 2.5–3 times stronger under the control of the Ubi promoter in the transgenic line than in wild-type males (*y¹w¹¹¹⁸* line) (*Figure 1B*, *Figure 1—figure supplement 3A*). Increased expression of MSL1 leads to the appearance of more MSL2, which is stabilized in a complex with MSL1. Similar results were obtained in females. In the absence of MSL2, the expression of MSL1 was lower in females compared to males (*Figure 1B*).

To compare the expression of the MSL1 variants in transgenic lines, we examined the levels of MSL1 expression in 2–3 day-old *msl-1⁻* females, which are homozygous for the transgenes expressing MSL1 variants. Immunoblot analysis showed that all MSL1 variants, with the exception of MSL1$^{\Delta1-85}$

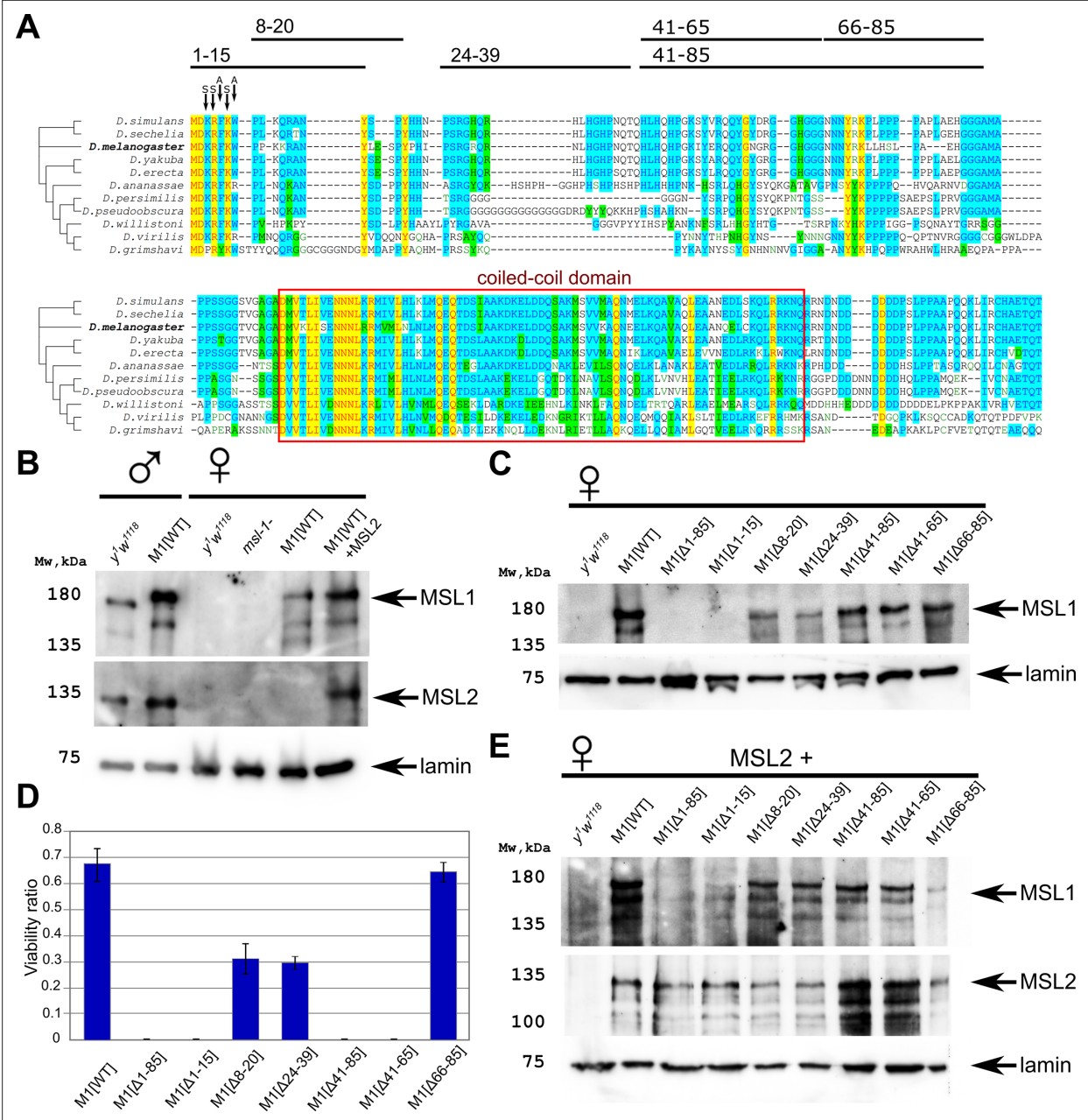

**Figure 1.** Testing the functional activity of male-specific lethal (MSL)1 mutants. (**A**) Schematic presentation of the MSL1 deletion proteins expressed in flies. Alignment of the N-terminal domain (1–207 aa) of MSL1 among Drosophilidae. (**B**) Immunoblot analysis of protein extracts prepared from adult males $y^1w^{1118}$ (control) and *msl-1⁻; Ubi:msl-1^{WT}/ Ubi:msl-1^{WT}(M1[WT])*, and from adult females $y^1w^{1118}$ (control), *⁻msl-1⁻ (msl-1^{L60}/msl-1^{y269})*, M1[WT] *(msl-1⁻; Ubi:msl-1^{WT}/Ubi:msl-1^{WT})*, and M1[WT]+MSL2 *(msl-1⁻; Ubi:msl-2^{WT}-FLAG / Ubi:msl-1^{WT})*. Immunoblot analysis was performed using anti-MSL1, anti-MSL2, and anti-lamin Dm0 (internal control) antibodies. (**C**) Immunoblot analysis of protein extracts prepared from adult females expressing different MSL1 protein variants (MSL1^{WT}, MSL1^{Δ1-85} MSL1^{Δ1-15}, MSL1^{Δ8-20} MSL1^{Δ24-39}, MSL1^{Δ41-85}, MSL1^{Δ41-65}, MSL1^{Δ66-85}) and $y^1w^{1118}$ females (control). Immunoblot analysis was performed using anti-MSL1, and anti-lamin Dm0 (internal control) antibodies. (**D**) Viability of males expressing MSL1 variants in the *msl-1*-null background (marked as M1[*]). Viability (as a relative percentage) of *msl-1⁻(msl-1^{L60}/msl-1^{y269}); Ubi:msl-1*/ Ubi:msl-1** males to *msl-1⁻; Ubi:msl-1*/Ubi:msl-1** females obtained in the progeny of crosses between females and males with the *msl-1⁻/ CyO, GFP; Ubi:msl-1*/Ubi:msl-1** genotype. *Ubi:msl-1** is any transgene expressing one of the tested MSL1 variants. The results are expressed as the mean of three independent crosses, with error bars showing standard deviations. (**E**) Immunoblot analysis of protein extracts prepared from adult female flies expressing MSL2-FLAG in combination with different MSL1 protein variants (MSL1^{WT}, MSL1^{Δ1-85} MSL1^{Δ1-15}, MSL1^{Δ8-20} MSL1^{Δ24-39}, MSL1^{Δ41-85}, MSL1^{Δ41-65}, MSL1^{Δ66-85}) and $y^1w^{1118}$ females (control). Immunoblot analysis was performed using anti-MSL1, anti-MSL2, and anti-lamin Dm0 (internal control) antibodies.

The online version of this article includes the following source data and figure supplement(s) for figure 1:

*Figure 1 continued on next page*

*Figure 1 continued*

**Source data 1.** Original files for the immunoblot blot analysis are in *Figure 1B, C and E*.

**Source data 2.** Files containing original immunoblots for *Figure 1B, C and E*, indicate the relevant bands.

**Source data 3.** The spreadsheet with numbers of males and females expressing male-specific lethal (MSL)1 variants in the *msl-1*-null background and obtained in three independent experiments.

**Figure supplement 1.** Alignment of the N-terminal region of male-specific lethal (MSL)1 among Diptera.

**Figure supplement 2.** Scheme of the transgenic construct used to express male-specific lethal (MSL)1 variants.

**Figure supplement 3.** Immunoblot analysis of M1[wt] flies.

**Figure supplement 3—source data 1.** Original files for the immunoblot blot analysis are in *Figure 1—figure supplement 3*.

**Figure supplement 3—source data 2.** Files containing original immunoblots for *Figure 1—figure supplement 3* indicating the relevant bands.

and MSL1$^{\Delta 1-15}$, were expressed in transgenic females at nearly equivalent levels (*Figure 1C*). Surprisingly, MSL1$^{\Delta 1-85}$ and MSL1$^{\Delta 1-15}$ could not be detected by immunoblotting, suggesting the instability of these MSL1 variants.

Homozygous *msl-1⁻* females remained viable, whereas males died during the late larva and early pupa stages (*Gorman et al., 1993*; *Palmer et al., 1993*). The transgenes, expressed MSL1$^{WT}$ and MSL1$^{\Delta 66-85}$, compensated for the viability of *msl-1⁻* males, indicating that these MSL1 variants are functional (*Figure 1D*, *Figure 1—source data 3*). In MSL1$^{\Delta 24-39}$ and MSL1$^{\Delta 8-20}$ lines, male viability was reduced suggesting that dosage compensation is partially affected. In the end, all other tested mutant MSL1 proteins (MSL1$^{\Delta 1-15}$, MSL1$^{\Delta 1-85}$, MSL1$^{\Delta 41-65}$, and MSL1$^{\Delta 41-85}$) failed to compensate for the *msl-1⁻* mutation in males (*Figure 1D*). Taken together, these results suggested that at least two regions (1–15 aa and 41–65 aa) in the N-terminus of MSL1 are critical for the functional activity of the MSL complex.

We then asked how MSL2 expression might affect the stability of MSL1 variants. To answer this question, we measured MSL1, and MSL2 proteins in *msl-1⁻*; *Ubi:msl-2$^{WT}$-FLAG/ Ubi:msl-1\** (*Figure 1E*). Again, immunoblotting showed that the amount of MSL1 was almost completely eliminated in females expressed either MSL1$^{\Delta 1-15}$ and MSL2, or MSL1$^{\Delta 1-85}$ and MSL2 (*Figure 1E*). All other MSL1 variants were present at the same level. Interestingly, there is no noticeable difference in the amount of MSL2 in all samples. Thus, the N-terminal 1–15 aa is essential for the stability of MSL1 independently of MSL2 presence.

## The N-terminus of MSL1 regulates the binding of the MSL complex to the X chromosome in females expressing MSL2

We next examined the extent to which deletions in the N-terminal region of MSL1 would affect the recruitment efficiency of the MSL complex to the male X chromosome. Immunostaining of polytene chromosomes from the salivary glands of *Drosophila* larvae allows the visualization of proteins on interphase chromatin and has been used extensively to study dosage compensation (*Dahlsveen et al., 2006*; *Demakova et al., 2003*; *Li et al., 2005*; *Lyman et al., 1997*; *Meller and Rattner, 2002*; *Palmer et al., 1994*). In the MSL1$^{\Delta 1-15}$, MSL1$^{\Delta 41-85}$, MSL1$^{\Delta 41-65}$, and MSL1$^{\Delta 1-85}$ mutant males, rare larvae surviving to 2–3 stages had the polytene chromosomes with poor morphology. For this reason, we used a previously described sensitive model system to study the factors required for the recruitment of the MSL complex to the X chromosome in females (*Chang and Kuroda, 1998*; *Li et al., 2008*; *Morra et al., 2011*; *Palmer et al., 1994*; *Zhou et al., 1995*). It was shown (*Tikhonova et al., 2019*), that MSL1 and MSL2 were specifically recruited to the X chromosome in transgenic females expressing MSL2$^{WT}$-FLAG under the control of the Ubi promoter inserted into attP-line 86Fb (*Ubi:msl-2$^{WT}$-FLAG*).

When the MSL2$^{WT}$-FLAG was expressed in the *msl-1⁻* background, we did not observe the binding of the MSL complex to the female X chromosome (data not shown). However, the binding of MSL1 and MSL2 to the X chromosome was detected in *msl-1⁻* females when MSL2$^{WT}$-FLAG was expressed simultaneously with either MSL1$^{WT}$ or MSL1$^{\Delta 66-85}$ (*Figure 2*, *Figure 2—figure supplement 1*). These results further supported that the 66–85-aa region is not essential for MSL1 function in dosage compensation.

The expression of all MSL1 variants with deletions of the first 15-aa region (MSL1$^{\Delta 1-15}$ and MSL1$^{\Delta 1-85}$) resulted in the near-complete loss of the MSL complex binding to the X chromosome in females (*Figure 2*, *Figure 2—figure supplement 1*). This result is expected as immunoblotting showed a

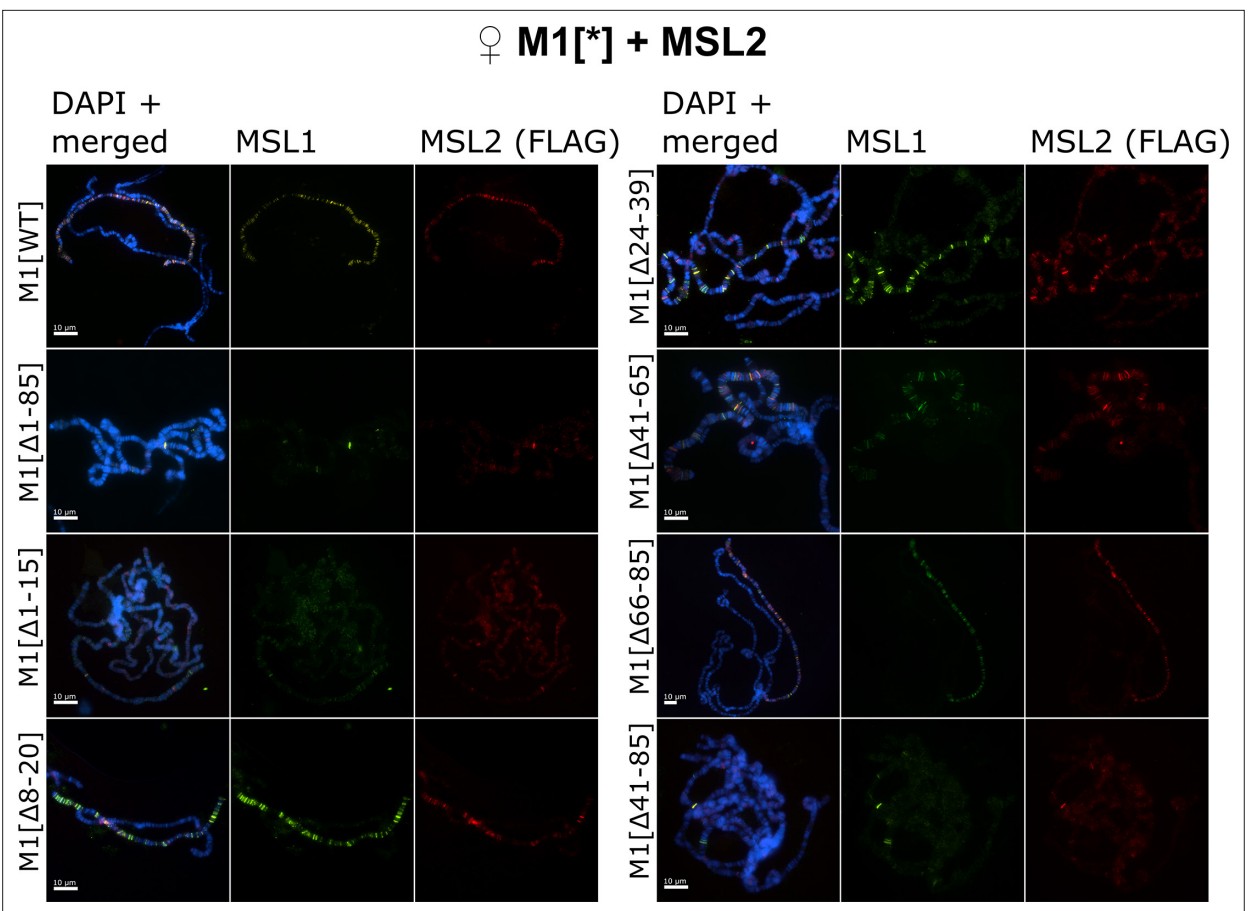

**Figure 2.** Testing the role of male-specific lethal (MSL)1 mutants in recruiting the MSL complex on the X chromosome in a female model system. Distribution of the MSL complex on the polytene chromosomes from third day female larvae expressing both MSL1* variants and the MSL2^WT-FLAG protein. Panels show the immunostaining of proteins using rabbit anti-MSL1 antibody (green) and mouse anti-FLAG antibody (red). DNA was stained with DAPI (blue). M1[WT](♀ *msl-1⁻*; *Ubi:msl-2^WT-FLAG/ Ubi:msl-1^WT*); M1[Δ1–85] (♀ *msl-1⁻*; *Ubi:msl-2^WT-FLAG/ Ubi:msl-1^Δ1-85*); M1[Δ1–15] (♀ *msl-1⁻*; *Ubi:msl-2^WT-FLAG/ Ubi:msl-1^Δ1-15*); M1[Δ8–20] (♀ *msl-1⁻*; *Ubi:msl-2^WT-FLAG/ Ubi:msl-1^Δ8-20*); M1[Δ24–39] (♀ *msl-1⁻*; *Ubi:msl-2^WT-FLAG/ Ubi:msl-1^Δ24-39*); M1[Δ41–85] (♀ *msl-1⁻*; *Ubi:msl-2^WT-FLAG/ Ubi:msl-1^Δ41-85*); M1[Δ41–65] (♀ *msl-1⁻*; *Ubi:msl-2^WT-FLAG/ Ubi:msl-1^Δ66-85*); M1[Δ66–85] (♀ *msl-1⁻*; *Ubi:msl-2^WT-FLAG/ Ubi:msl-1^Δ66-85*).

The online version of this article includes the following source data and figure supplement(s) for figure 2:

**Source data 1.** Raw files with images of polytene chromosomes.

**Figure supplement 1.** Testing the role of male-specific lethal (MSL)1 mutants in recruiting the MSL complex on the X chromosome in a female model system.

**Figure supplement 2.** Cytological localization of male-specific lethal (MSL)1 variants and MSL2 on the straightened polytene chromosomes of the M1[Δ1–15] (♀ *msl-1⁻*; *Ubi:msl-2^WT-FLAG/ Ubi:msl-1^Δ1-15*); M1[Δ8–20] (♀ *msl-1⁻*; *Ubi:msl-2^WT-FLAG/ Ubi:msl-1^Δ8-20*); M1[Δ24–39] (♀ *msl-1⁻*; *Ubi:msl-2^WT-FLAG/ Ubi:msl-1^Δ24-39*); M1[Δ41–85] (♀ *msl-1⁻*; *Ubi:msl-2^WT-FLAG/ Ubi:msl-1^Δ41-85*); M1[WT] (♀ *msl-1⁻*; *Ubi:msl-2^WT-FLAG/ Ubi:msl-1^WT*).

strong decrease of MSL1 level in MSL1^Δ1-15/MSL2^WT and MSL1^Δ1-85/MSL2^WT females (*Figure 1E*). Previous studies (*Dahlsveen et al., 2006*; *Demakova et al., 2003*; *Lyman et al., 1997*; *Meller and Rattner, 2002*; *Palmer et al., 1994*) showed that the inactivation of MSL3, MLE, or roX RNAs resulted in the recruitment of the MSL complex to a small proportion of bands that correspond to the main HAS, including regions of the *roX1* (3 F) and *roX2* (10 C) genes. In polytene chromosomes in the MSL1^Δ1-15 and MSL1^Δ1-85 lines, we found only two intense and several faint MSL1/MSL2 bands (*Figure 2—figure supplement 2*). Interestingly, additional MSL1/MSL2 bands were identified on the polytene autosomes, suggesting that the MSL proteins began to bind to lower affinity sites on autosomes (*Figure 2—figure supplement 1*). Thus, the deletion of the N terminal 15 aa affects not only the stability of MSL1 but also the specificity of MSL binding to the X chromosome.

The deletion of the 41–85 aa or 41–65 aa regions did not affect the stability of MSL1, but at the same time significantly reduced the binding of MSL proteins to the X chromosome. MSL1$^{\Delta41-85}$ and MSL2 bind to several bands on the X chromosome, many of which can also be identified as weak bands in the MSL1$^{\Delta1-15}$ and MSL1$^{\Delta1-85}$ lines (*Figure 2*, *Figure 2—figure supplements 1 and 2*). In comparison with MSL1$^{\Delta1-15}$ and MSL1$^{\Delta1-85}$, MSL proteins bind more efficiently with the MSL1$^{\Delta41-85}$ and MSL1$^{\Delta41-65}$ polytene chromosomes. However, the binding of the MSL complex to the X chromosome in MSL1 mutants was much weaker than previously suggested for a minimal MSL complex lacking either MSL3 or MLE (*Dahlsveen et al., 2006*). As the 66–85-aa deletion did not affect the activity of MSL1, we concluded that the 41–65-aa region is critical for MSL1 function.

Deletion of 24–39 aa, which separates the 1–15 aa and 41–65 aa regions in MSL1, only moderately affects the binding of the MSL2 and MSL1$^{\Delta24-39}$ proteins on polytene chromosomes (*Figure 2*, *Figure 2—figure supplement 1*). This observation indicates that the 1–15 aa and 41–65 aa regions have at least partially independent functions in recruiting of the MSL complex to the X chromosome in males. In MSL1$^{\Delta8-20}$ female larvae patterns of MSL1 and MSL2 binding to polytene chromosome are the same as in larvae expressed MSL1$^{\Delta24-39}$ (*Figure 2*, *Figure 2—figure supplements 1 and 2*). Thus, a deletion from 8 aa up to 20 aa only partially affects the functions of MSL1, in contrast to the complete inactivation of this protein by deleting the terminal 15 aa (MSL1$^{\Delta1-15}$).

## The N terminal 3-7 amino acids of MSL1 are critical for dosage compensation

Our results showed that MSL1$^{\Delta1-15}$ is unstable, while MSL1$^{\Delta8-20}$ only partially affects the specific recruitment of MSL to the X chromosome suggesting that the most terminal amino-acids are essential for functions. To test the role of the N-terminal amino acids (3-7) in the stability and functions of MSL1, we made several amino acids substitutions in this region. The N-terminal region contains well-conserved basic and aromatic amino acid residues (*Figure 1A*). We used data about the previously characterized MSL1 mutants (*Li et al., 2005*) in those three conserved basic amino acids, lysine 3, arginine 4, and lysine 6, or two conserved aromatic amino acids, phenylalanine 5 and tryptophan 7, were replaced by alanine. In our work, we replaced basic amino acids with serine, and aromatic amino acids – with alanine or glycine (*Figure 3A*). In result we obtained three different transgenes: MSL1$^{3S}$ (K3S R4S K6S), MSL1$^{2A}$ (F5A W7A), and MSL1$^{GS}$ (K3S R4S F5G K6S W7G).

To evaluate the functional effects of N-terminal amino acids of the MSL1 protein in vivo, we created transgenic flies expressing mutants MSL1$^{2A}$, MSL1$^{3S,}$ and MSL1$^{GS}$ under the control of the Ubi promoter. All MSL1 variants were targeted by the HAx3 epitope at the C-termini (*Figure 3A*). MSL1$^{2A}$ and MSL1$^{3S}$ variants were expressed at levels comparable to MSL1$^{WT}$ in 2–3 day-old *msl-1$^-$* females that are homozygous for the MSL1 transgenes (*Figure 3B and C*). At the same time, there was an approximately 1.5–2-fold decrease in the MSL1$^{GS}$ protein compared to MSL1$^{WT}$.

The transgenes, expressed MSL1$^{2A}$ and MSL1$^{3S}$, almost completely compensated for the viability of *msl-1$^-$* males, indicating that these MSL1 variants are functional (*Figure 3D*, *Figure 3—source data 3*). In the *msl-1$^-$* line expressing MSL1$^{GS}$, the viability of males was strongly reduced. At the same time, increasing in MSL2 and a simultaneous decreasing in MSL1$^{GS}$ expression partially restore the viability of MSL1$^{GS}$/MSL2 males (*Figure 3E*, *Figure 3—source data 4*). In a similar way, the viability of MSL1$^{\Delta8-20}$/MSL2 males also is close to wild-type, but MSL1$^{\Delta41-65}$/MSL2 and MSL1$^{\Delta1-15}$/MSL2 males are still lethal.

According to the results obtained on males, MSL1$^{2A}$ and MSL1$^{3S}$ showed binding patterns like MSL1$^{WT}$ on the larva polytene chromosome, while the MSL1$^{GS}$ protein failed to effectively bind on the X chromosome in complex with MSL2 (*Figure 3F*, *Figure 3—figure supplements 1 and 2*). There are also binding sites for only MSL2 or MSL1$^{GS}$ on the autosomes and the X chromosome. Taken the obtained results suggest that the N-terminal amino acids 3–7 additively contribute to specific recruitment of MSL on the X chromosome.

Next, we asked whether MSL1$^{\Delta1-15}$, MSL1$^{\Delta8-20}$, MSL1$^{\Delta24-39}$, MSL1$^{\Delta41-65}$, MSL1$^{GS}$, MSL1$^{\Delta1-85}$, MSL1$^{\Delta41-85}$ variants exhibit similar stability when expressed in *Drosophila* male-like S2 cells. All MSL1 variants targeted by HAx3 were co-expressed with MSL2-FLAG in S2 cells (*Figure 4A*). MSL1$^{WT}$ was stable and effectively immunoprecipitated with MSL2 (*Figure 4B*). Similar results were obtained with MSL1$^{\Delta8-20}$, MSL1$^{\Delta24-39}$, MSL1$^{\Delta41-85}$, MSL1$^{2A}$, MSL1$^{3S}$, and MSL1$^{GS}$ (*Figure 4B*). Like in *Drosophila*, MSL1$^{\Delta1-15}$ and MSL1$^{\Delta1-85}$ were unstable and expressed at a very low level. These results confirm that the specific combinations of amino-acids at the N-terminus are required for the stability of the MSL1 protein.

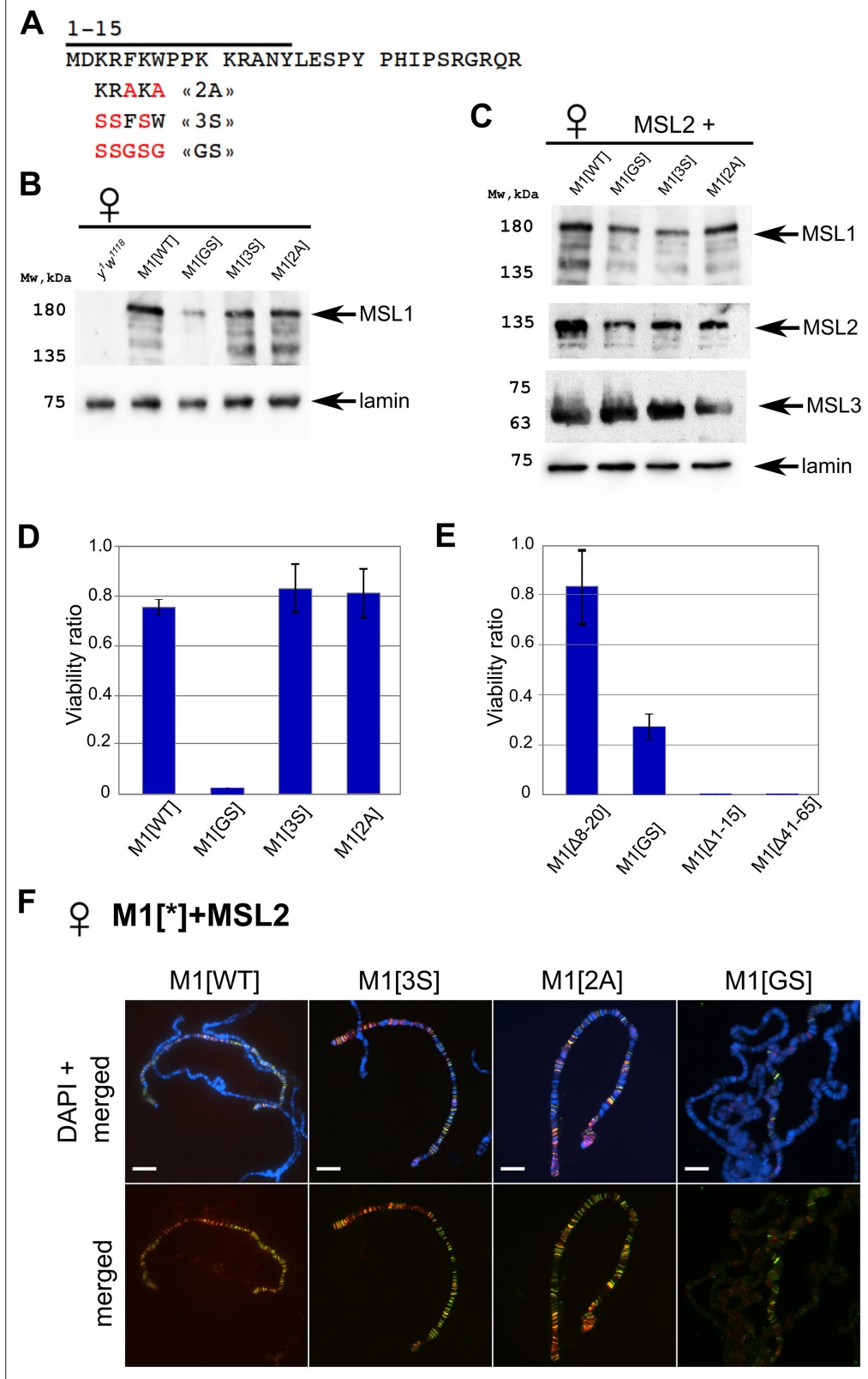

**Figure 3.** Testing the functional role of N-terminal amino acids 3–7 of male-specific lethal (MSL)1. (**A**) Schematic presentation of the mutations at the N-terminus of MSL1. (**B**) Immunoblot analysis of protein extracts prepared from adult female flies expressing different MSL1* variants in the *msl-1⁻* (*msl-1^{L60}/msl-1^{γ269}*) background (M1[WT]: MSL1^{WT}; M1[2 A]: MSL1^{F5A W7A}, M1[3 S]: MSL1^{K3S R4S K6S}; M1[GS]: MSL1^{K3S R4S F5G K6S W7G}). (**C**) Immunoblot analysis of

*Figure 3 continued on next page*

*Figure 3 continued*

protein extracts prepared from adult female flies expressing MSL2-FLAG in combination with different MSL1* variants. Immunoblot analysis was performed using anti-MSL1, anti-MSL2, anti-MSL3, and anti-lamin Dm0 (internal control) antibodies. (**D**) Viability of males expressing MSL1 variants (marked as M1[*]) in the *msl-1⁻* (*msl-1^{L60}/msl-1^{Y269}*) background. Viability (as a relative percentage) of *msl-1⁻; Ubi:msl-1*/ Ubi:msl-1** males to *msl-1⁻; Ubi:msl-1*/Ubi:msl-1** females obtained in the progeny of a cross between females and males with *msl-1⁻/ CyO, GFP; Ubi:msl-1*/Ubi:msl-1** genotype. *Ubi:msl-1** is any transgene expressing one of the tested MSL1* variants. (**E**) Viability of males expressing MSL2^{WT} and MSL1* variants (marked as M1[*]) in the *msl-1⁻* background. Viability (as a relative percentage) of *msl-1⁻; Ubi:msl-2^{WT}-FLAG /Ubi:msl-1** males to *msl-1⁻; Ubi:msl-2^{WT}-FLAG /Ubi:msl-1** females obtained in the progeny of cross between males *msl-1⁻/ CyO, GFP; Ubi:msl-1*/Ubi:msl-1** and females *msl-1⁻; Ubi:msl-2/Ubi:msl-2*. The results are expressed as the mean of three independent crosses, with error bars showing standard deviations. (**F**) Distribution of the MSL complex on the polytene chromosomes from third day female larvae expressing both MSL1* variants and the MSL2-FLAG protein. Panels show the merged results of immunostaining of MSL1 (green, rabbit anti-MSL1 antibody) and MSL2 (red, mouse anti-FLAG antibody). DNA was stained with DAPI (blue). Scale bar indicates 10 μm.

The online version of this article includes the following source data and figure supplement(s) for figure 3:

**Source data 1.** Original files for the immunoblot blot analysis in *Figure 3B and C*.

**Source data 2.** Files containing original immunoblots for *Figure 3B and C* indicating the relevant bands.

**Source data 3.** The spreadsheet with numbers of males and females expressing male-specific lethal (MSL)1 variants in the *msl-1*-null background and obtained in three independent experiments.

**Source data 4.** The spreadsheet with numbers of males and females expressing MSL2^{WT} and male-specific lethal (MSL)1 variants in the *msl-1*-null background and obtained in three independent experiments.

**Figure supplement 1.** Distribution of the male-specific lethal (MSL) complex on the polytene chromosomes from third day female larvae expressing both MSL1* variants and the MSL2-FLAG protein.

**Figure supplement 2.** Cytological localization of male-specific lethal (MSL)1 variants and MSL2 on the straightened polytene chromosomes of the M1[WT] (♀ *msl-1⁻; Ubi:msl-2^{WT}-FLAG/ Ubi:msl-1^{WT}*); M1[2 A] (♀ *msl-1⁻; Ubi:msl-2^{WT}-FLAG/ Ubi:msl-1^{2A}*); M1[3 S] (♀ *msl-1⁻; Ubi:msl-2^{WT}-FLAG/ Ubi:msl-1^{3S}*); M1[GS] (♀ *msl-1⁻; Ubi:msl-2^{WT}-FLAG/ Ubi:msl-1^{GS}*). Panels show the immunostaining of proteins using mouse anti-HA antibody (green) and rabbit anti-MSL2 antibody (red). DNA was stained with DAPI (blue).

The results in S2 cells showed that MSL1^{Δ41-85} and MSL1^{GS} can co-immunoprecipitate with MSL2, as does MSL1^{WT} (*Figure 4B*). These results suggest that mutant MSL1^{Δ41-65} and MSL1^{GS} may be part of the MSL complex. Since S2 cells contain an endogenous MSL1 protein, it seems likely that the mutated MSL1 variants form heterodimers with wild-type ones. We then asked if these mutant protein variants could interact with endogenous MSL1 for specific recruitment to the X chromosome. To distinguish between WT and MSL1 mutants, we generated transgenic lines expressing MSL1^{Δ41-85}-HA or MSL1^{GS}-HA tagged with the HAx3 epitope at the *msl-1⁺* background. In transgenic males and females expressed MSL2 and MSL1 variants in the heterozygous state, we observed complete co-localization between MSL2 and MSL1 variants marked by HAx3 epitope (+). Thus, in the presence of wild-type MSL1, MSL1^{GS} and MSL1^{Δ41-85} variants enter the MSL complex, in which they specifically bind to the male X chromosome. These results suggest that MSL1^{GS} and MSL1^{Δ41-85} effectively dimerize with the wild-type MSL1 via the coiled-coil domain.

## The mutants MSL1^{GS} and MSL1^{Δ41-85} disrupt the recruitment of the MSL complex to HAS

For a more complete understanding of the role played by the N-terminal region of MSL1 in the recruitment of the MSL complex to the X chromosome, we tested the binding of the MSL1, MSL2, and MSL3 proteins by chromatin immunoprecipitation followed by next-generation sequencing (ChIP-seq) with a standard protocol for chromatin fragmentation based on sonication. We obtained MSL protein profiles from two biological replicates of 3-day-old adult males and females.

As the binding profiles of the MSL complex have been well-studied in cell lines, we compared MSL1, MSL2, and MSL3 binding efficiency at the HAS (*Figure 5—figure supplement 1*) with previously published data from S2 cells (*Schauer et al., 2017*; *Straub et al., 2013*). The ChIP-seq data from adult *y¹w^{1118}* (WT) males showed MSL binding to approximately 90% of the previously identified HAS from three different studies (*Alekseyenko et al., 2008*; *Ramírez et al., 2015*; *Straub et al., 2008*;

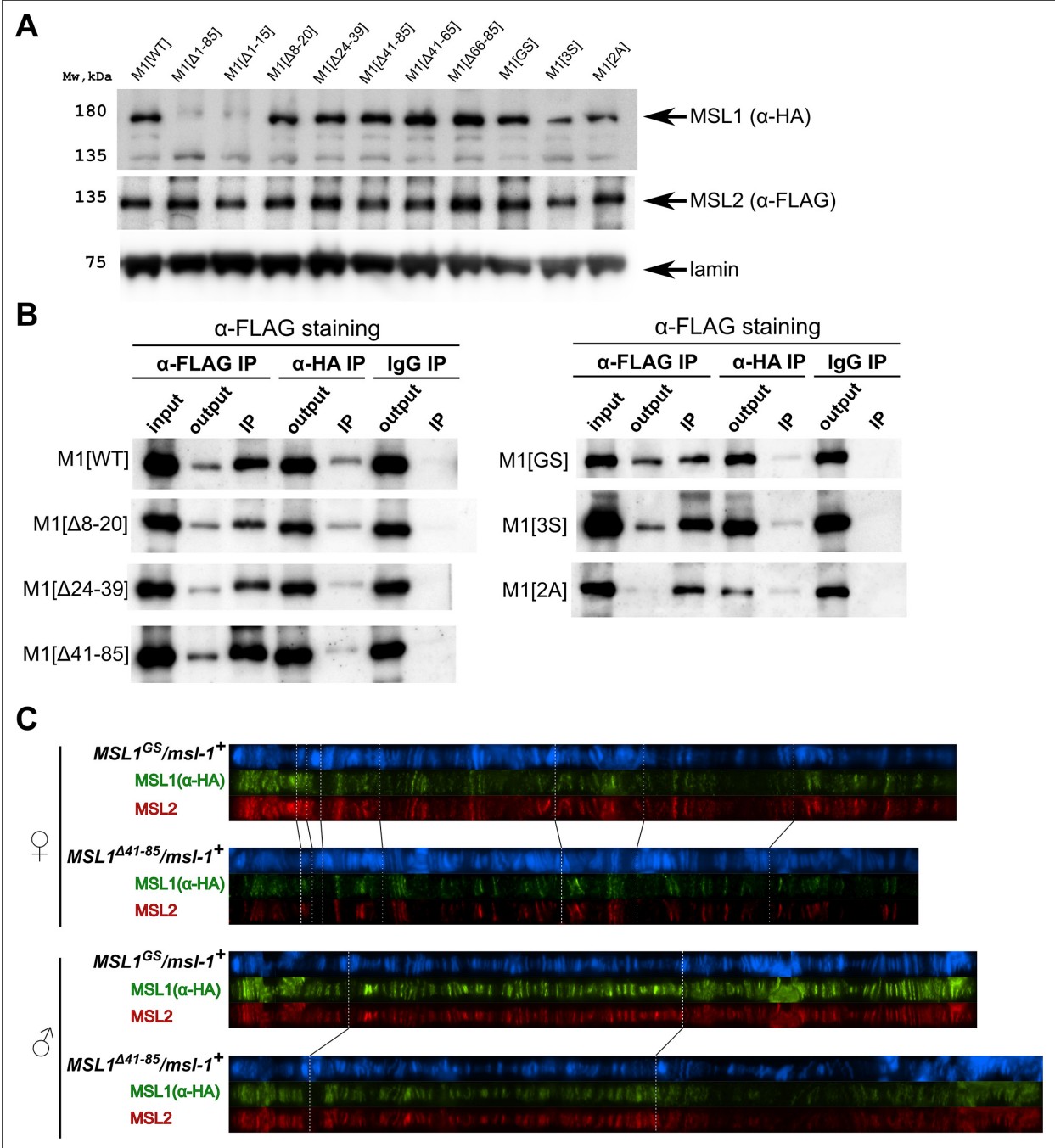

**Figure 4.** Testing the functional activity of mutant variants of the male-specific lethal (MSL)1 protein. (**A**). Immunoblot analysis of protein extracts prepared from S2 cells expressing MSL2-FLAG and different MSL1-HAx3 protein variants (M1[WT]: MSL1$^{WT}$; M1[Δ*]: MSL1$^{Δ*}$; M1[2 A]: MSL1$^{2A}$, M1[3 S]: MSL1$^{3S}$; M1[GS]: MSL1$^{GS}$). Immunoblot analysis was performed using anti-HA (MSL1-HAx3 variants), anti-FLAG (MSL2-FLAG), and anti-lamin Dm0 (internal control) antibodies. (**B**) Protein extracts from *Drosophila* S2 cells co-transfected with different MSL1-HAx3 variants and MSL2-FLAG were immunoprecipitated with antibodies against FLAG or HA or nonspecific mouse IgG as a negative control. The immunoprecipitates were analyzed by immunoblotting for the presence of FLAG-tagged proteins in immunoprecipitated samples. (**C**) Distribution of the MSL1*-HA and MSL2-FLAG on the polytene chromosomes from third day female or male larvae expressing one of MSL1*-HA variants and MSL2-FLAG at the wild-type background (*msl-1+*; *Ubi:msl-2$^{WT}$-FLAG /Ubi:msl-1**). Panels show the merged results of immunostaining of MSL1*-HA (green, mouse anti-HA antibody) and MSL2 (red, rabbit anti-MSL2 antibody). DNA was stained with DAPI (blue).

The online version of this article includes the following source data for figure 4:

**Source data 1.** Original files for the immunoblot blot analysis are in *Figure 4A and B*.

**Source data 2.** Files containing original immunoblots for *Figure 4A and B* indicating the relevant bands.

*Figure 5—source data 1*). As a control, the MSL1 and MSL3 proteins bind to autosomal sites but not to HAS in females (*Figure 5—figure supplement 1A*). These results indicate that most HAS are common between S2 cells and adult males.

Next, we compared the binding profiles of MSL1, MSL2, and MSL3 proteins obtained in *msl-1⁻; Ubi:msl-2^{WT}-FLAG/Ubi:msl-1^{WT}* females (M1[WT]+MSL2) and *msl-1⁻; Ubi:msl-1^{WT}/Ubi:msl-1^{WT}* males (M1[WT]) (*Figure 5—figure supplement 2A*). We found a good correlation between the obtained profiles, demonstrating that MSL2 expressed in females induces the binding of the MSL complex to the same HAS regions. We observed stronger binding of the MSL1, MSL2, and MSL3 proteins in M1[WT] (*msl-1⁻; Ubi:msl-1^{WT}/Ubi:msl-1^{WT}*) males compared with those in control *y¹w¹¹¹⁸* males (*Figure 5—figure supplement 1B*, *Figure 5—figure supplement 2A*), which might be a consequence of the higher concentrations of MSL proteins in the transgenic line due to being expressed under the control of the Ubi promoter (*Figure 1—figure supplement 3A*). In M1[WT]+MSL2 females, the MSL proteins also bind to most HAS, but with lower efficiency (*Figure 5—figure supplement 2A, B, C*). These results are consistent with previous studies demonstrating that the ectopic expression of MSL2 in females is sufficient to trigger the correct targeting of MSL proteins to the X chromosome (*Valsecchi et al., 2018*).

To test the role played by the N-terminal region of MSL1, we compared the binding of MSL proteins in females expressing MSL2 combined with MSL1^{WT}, MSL1^{Δ1-15}, MSL1^{Δ41-85}, and MSL1^{GS} (*Figure 5A*). As shown above, MSL2 and MSL1^{WT} effectively bind to most HAS in females. MSL3 was associated with extended regions surrounding the HAS, consistent with previously reported data (*Larschan et al., 2007*; *Sural et al., 2008*). In MSL1^{Δ1-15}, MSL1^{Δ41-85}, and MSL1^{GS} flies binding of MSL1, MSL2, and MSL3 to X chromosomal HAS is strongly reduced (*Figure 5A-E*, *Figure 5—figure supplement 3A*). At the same time, we noticed that there are HAS in which MSL2 continues to bind effectively when MSL1 in MSL1^{Δ1-15}, MSL1^{Δ41-85}, and MSL1^{GS} flies are strongly reduced (*Figure 5—figure supplement 3B*).

To investigate observed differences in localization of the MSL proteins at various genomic sites in female fly lines, we calculated the average logFC between normalized (RPKM) test signals and nonspecific IgG signals in regions where MSL1 and MSL2 colocalized in MSL1^{WT} line on X chromosome (see Methods). Resulting list of peaks is mostly associated with HAS (217 out of 220 in total) and is highly enriched by the MSL1, MSL2, and MSL3 proteins in MSL1^{WT} flies, suggesting the binding of the complete MSL complex. As a result of hierarchical clustering, these regions were divided into two groups based on the protein binding profiles of MSL1, MSL2, and MSL3. Cluster 1 ('MSL2 Sensitive,' N=133) contains sites to which proteins do not bind during expression of MSL1 mutants. Cluster 2 ('MSL2 stable,' N=87) includes sites to which MSL2 continues to bind even in the presence of MSL1 mutations (see Methods, *Figure 6A*, *Figure 6—figure supplement 1*, *Figure 6—source data 1*). In cluster 2 regions, in addition to MSL2, the mutant form of MSL1^{GS} was also enriched. We decided to test whether the stability of MSL2 in the cluster 2 is associated with the presence of PionX sites, to which the MSL2 protein binds most effectively in vitro (*Villa et al., 2016*). Indeed, PionX sites made up a larger percentage of the peaks of the cluster 2 (17 out of 87), in contrast to cluster 1 (7 out of 133). Additionally, we tested the strength of MSL1 and MSL2 signals from female fly lines in peaks associated with PionX sites in comparison with all other regions from the cluster 2 (*Figure 6B*). As expected, we found that the MSL2 signal was significantly higher in the group of PionX sites in fly lines with the mutation (MSL1^{Δ1-15}, MSL1^{Δ41-85}, MSL1^{GS}; Wilcoxon signed-rank test, p-value: 0.034, 7.3e-5, 2.2e-5, respectively), which was not noted for the MSL1^{WT} (Wilcoxon signed-rank test, p-value: 0.71). Similar situation was observed for MSL1, whose signal at PionX sites was significantly greater for the MSL1^{GS} females (Wilcoxon signed-rank test, p-value: 0.053) than for the MSL1^{WT} females (Wilcoxon signed-rank test, p-value: 0.35). These results suggest that some of the PionX sites may be bound independently by the MSL2 protein.

Also, we analyzed regions with MSL1 alone peaks (no colocalization with MSL2) on all chromosomes (*Figure 6—figure supplement 2*). Previously, it was shown that MSL1 binds to promoters independently of the MSL complex (*Chlamydas et al., 2016*; *Hallacli et al., 2012*; *Straub et al., 2013*). Accordingly, MSL2 binding was weak or absent at all such sites. These sites are predominantly located in the promoter regions and MSL1^{GS} binds to all sites with the same efficiency as MSL1^{WT}, confirming that this mutant protein retains its functionality at the sites outside of HAS. The MSL1^{Δ1-15} and MSL1^{Δ41-85} are not associated with these sites. The absence of MSL1^{Δ1-15} can be explained by the low stability of this mutant protein. As MSL1^{Δ41-85} is expressed more strongly than MSL1^{GS}, it seems

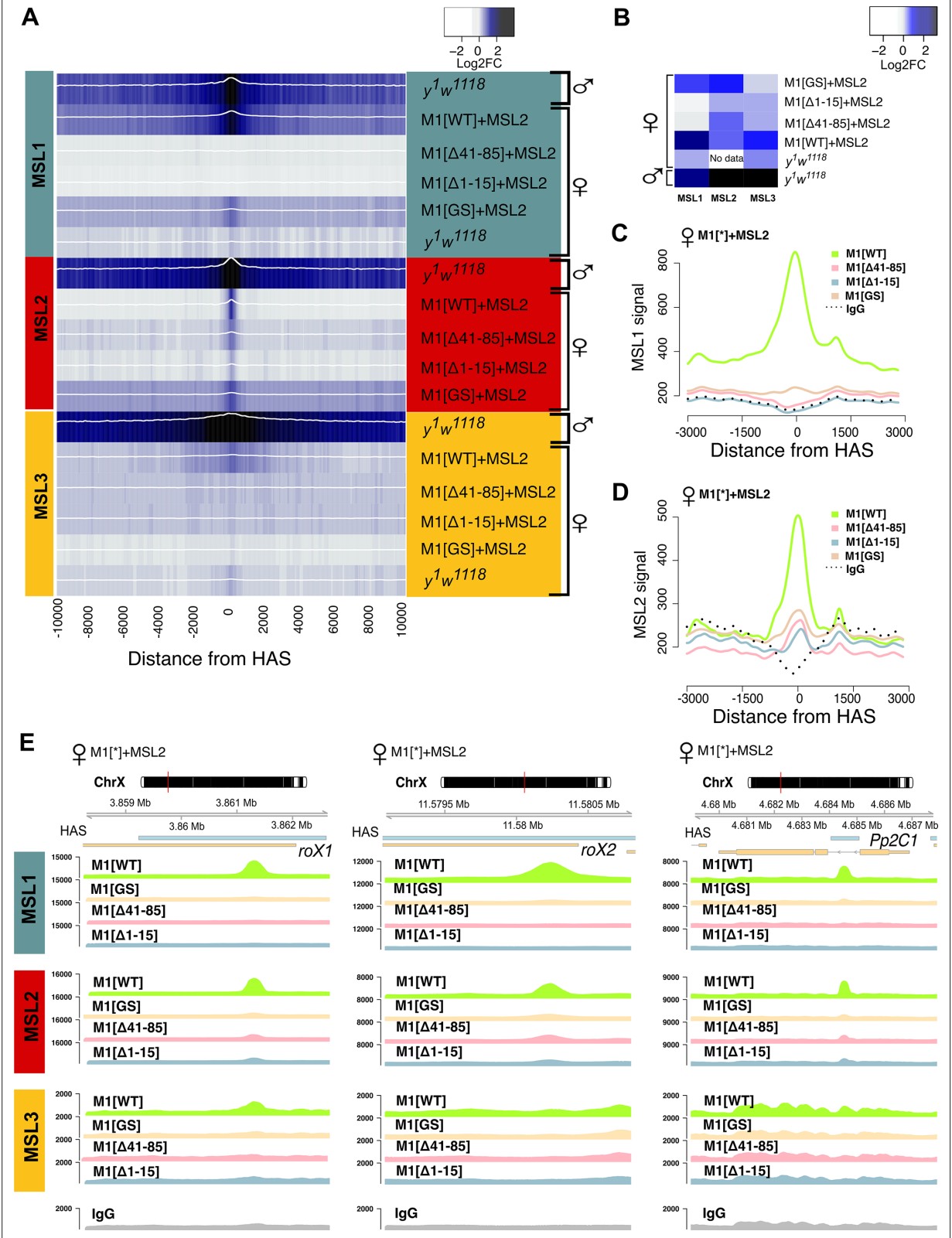

**Figure 5.** Role of the N-terminal regions in the recruitment of the dosage compensation complex to the high-affinity 'entry' sites (HAS). To study the functional role of the N-terminal region of male-specific lethal (MSL)1 for the recruitment of the dosage compensation complex, we compared the profiles of MSL1, MSL2, and MSL3 binding in 2–3 day females expressing MSL2 and one of four MSL1 variants (MSL1$^{WT}$, MSL1$^{GS}$, MSL1$^{\Delta1-15}$, MSL1$^{\Delta41-85}$). $y^1w^{1118}$ (wild-type males and females), ♀M1[WT]+MSL2 (*msl-1⁻*; *Ubi:msl-2$^{WT}$-FLAG/Ubi:msl-1$^{WT}$* females), ♀M1[Δ41–85]+MSL2 (*msl-1⁻*; *Ubi:msl-2$^{WT}$-FLAG/*

*Figure 5 continued*

Ubi:msl-1$^{\Delta41-85}$ females), ♀M1[Δ1–15]+MSL2 (msl-1⁻; Ubi:msl-2$^{WT}$-FLAG/Ubi:msl-1$^{\Delta1-15}$ females), ♀M1[GS]+MSL2 (msl-1⁻; Ubi:msl-2$^{WT}$-FLAG/Ubi:msl-1$^{GS}$ females). (**A**) Average log fold-change between normalized (RPKM) test signals and nonspecific IgG signals in HAS (see Materials and methods). Average log fold-change was calculated after smoothing signals using the Daniell kernel with kernel size 50 and the addition of a pseudocount. Next, the tracks were aggregated using 100 bp bins. (**B**) Heatmap showing average log fold-change between normalized (RPKM) test signals and nonspecific IgG signals in region ±500 bps from HAS centers. (**C and D**) Average signal (RPKM) for (**C**) MSL1 protein and (**D**) MSL2 protein in different fly lines at HAS (see Methods). (**E**) Enrichment of MSL1, MSL2, and MSL3 signals associated with the X-linked genes encoding *Pp2C1* and the non-coding RNAs *roX1* and *roX2* in females expressing MSL2 and one of four MSL1 variants (MSL1$^{WT}$, MSL1$^{GS}$, MSL1$^{\Delta1-15}$, MSL1$^{\Delta41-85}$).

The online version of this article includes the following source data and figure supplement(s) for figure 5:

**Source data 1.** A list of high-affinity 'entry' site (HAS) regions.

**Figure supplement 1.** Comparison of MSL1, MSL2 and MSL3 binding in the flies of $y^1w^{1118}$ and M1[wt] lines.

**Figure supplement 2.** Comparisons of the binding patterns of male-specific lethal (MSL) proteins between 2–3 day adult males and females expressing MSL2.

**Figure supplement 3.** Examples of male-specific lethal (MSL)1, MSL2, and MSL3 signal enrichment associated with X chromosomal high-affinity 'entry' sites (HAS) in M1[WT] (♀ msl-1⁻; Ubi:msl-2$^{WT}$-FLAG/ Ubi:msl-1$^{WT}$), M1[GS] (♀ msl-1⁻; Ubi:msl-2$^{WT}$-FLAG/ Ubi:msl-1$^{GS}$), M1[Δ41–85] (♀ msl-1⁻; Ubi:msl-2$^{WT}$-FLAG/ Ubi:msl-1$^{\Delta41-85}$), M1[Δ1–15] (♀ msl-1⁻; Ubi:msl-2$^{WT}$-FLAG/ Ubi:msl-1$^{\Delta1-15}$) females.

likely that the 41–85 aa region facilitates the interaction of MSL1 with transcription factors associated with these promoters.

## Two regions in the N-terminal domain are required for interaction with roX2 RNA

Our results suggest that MSL1$^{GS}$ and MSL1$^{\Delta41-85}$ can interact with wild-type MSL1 and form the complex. Previously it was suggested that the N-terminal part of MSL1 (191 aa) contributes to interaction with roX RNAs (*Müller et al., 2020*). It can be assumed that the ineffectiveness of MSL1 variants in dosage compensation is associated with their inability to interact with roX RNAs. To test this hypothesis, we obtained transgenic *msl-1⁻* (*msl-1$^{L60}$/msl-1$^{γ269}$*) females expressed either of MSL1 variants (MSL1$^{WT}$-HA, MSL1$^{GS}$-HA, and MSL1$^{\Delta41-85}$-HA) and roX2 RNA. To express roX2, we used the previously obtained *roX2* gene under the control of the Ubi promoter in the 86Fb site (*Tikhonova et al., 2022b*). The *roX2* transgene was expressed at the same level in all females expressed different MSL1 variants (*Figure 7A*).

The total extracts obtained from the 2–3 days old adult females (*Figure 7*) were used for immunoprecipitation with MSL1 antibodies. In all cases antibodies efficiently precipitate the tested MSL1 variants (*Figure 7B*). *roX2* RNA was detected (*Figure 7C*) in precipitate obtained from MSL1$^{WT}$ extract (positive control) and was not found in precipitate obtained from *msl-1⁻* extract (negative control). In the case of MSL1$^{GS}$ and MSL1$^{\Delta41-85}$ precipitates, *roX2* RNA was not detected as in the negative control. The obtained results suggest that MSL1$^{WT}$ interacts with roX2 in contrast to the MSL1$^{GS}$ and MSL1$^{\Delta41-85}$ mutants. These results are consistent with the hypothesis that failure to interact with roX RNAs is the primary cause of functional inactivation of MSL1$^{GS}$ and MSL1$^{\Delta41-85}$.

## Discussion

This study focused on examining the role played by the MSL1 protein in recruiting the MSL complex in *Drosophila* to HAS. By UV-XL assay, it was shown that the N-terminal 191 aa of MSL1 and MSL2 are direct roX2 interactors (*Müller et al., 2020*). Here, we found that the 15 N-terminal aa are critical for the stability of the MSL1 protein. The substitution of the N-terminal amino-acids 3–7 (KRFKW) on the SSGSG in MSL1$^{GS}$ slightly reduces the stability of the mutant protein and strongly affects the interaction of MSL1$^{GS}$ with roX2. The 41–85 aa are also critical for MSL1 function and interaction with roX2 RNA in vivo. Both regions are separated by a linker that is less essential for MSL1 activity.

As functions of roX1 and roX2 are redundant (*Meller and Rattner, 2002*; *Stuckenholz et al., 2003*), it seems likely that roX1 interacts with the same N-terminal regions of MSL1 as roX2. Further study is required to understand the mechanisms for how N-terminal sequences of MSL1 determine the highly specific interaction with roX RNAs. In accordance with the key role played in the recruitment of the MSL complex to the male X chromosome, the N-terminal 15 amino acids of MSL1 are highly conserved among Drosophilidae. However, MSL1 orthologs in other families of Diptera have completely different

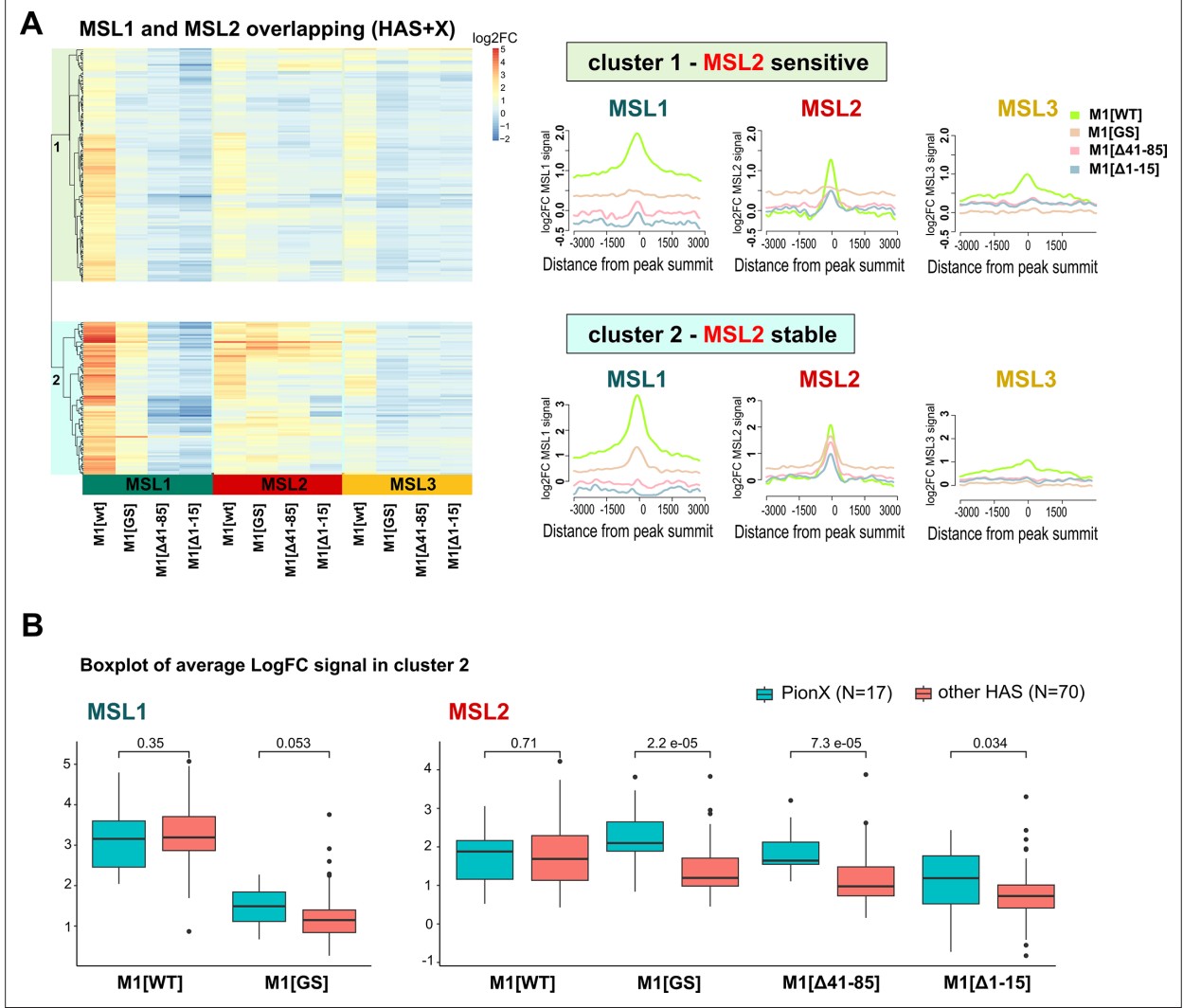

**Figure 6.** Male-specific lethal (MSL)1 shows different scenarios of binding upon N-terminal region modification. Regions clusterization with MSL1∩MSL2 overlapping peaks according to MSLs binding in different female fly lines expressing MSL2 and one of four MSL1 variants (MSL1$^{WT}$, MSL1$^{GS}$, MSL1$^{\Delta41-85}$, MSL1$^{\Delta1-15}$) (see Materials and methods). M1[WT] (*msl-1−; Ubi:msl-2$^{WT}$-FLAG/Ubi:msl-1$^{WT}$* females), M1[Δ41–85] (*msl-1−; Ubi:msl-2$^{WT}$-FLAG/Ubi:msl-1$^{\Delta41-85}$* females), M1[GS] (*msl-1−; Ubi:msl-2$^{WT}$-FLAG/Ubi:msl-1$^{GS}$* females), M1[Δ1–15] (*msl-1−; Ubi:msl-2$^{WT}$-FLAG/Ubi:msl-1$^{\Delta1-15}$* females). (**A**) Left - Heatmap of average logFC between MSL and nonspecific IgG signal in female fly lines, hierarchical clusterization, and cluster subdivision is drawn on the left corner. Right - Average logFC profiles of MSL signal in different clusters for female fly lines. Average log fold-change was calculated after smoothing signals using the Daniell kernel with kernel size 100 and the addition of a pseudocount. (**B**) Boxplot of average LogFC signal of MSL1 and MSL2 in cluster 2. All sites in cluster 2 were subdivided into PionX sites and other HAS, and the strength of MSL1 and MSL2 signals was tested.

The online version of this article includes the following source data and figure supplement(s) for figure 6:

**Source data 1.** A list of unique male-specific lethal (MSL)1 peaks for females M1[WT]+MSL2 fly line (*msl-1−; Ubi:msl-2$^{WT}$-FLAG/Ubi:msl-1$^{WT}$*).

**Figure supplement 1.** Examples of male-specific lethal (MSL)1, MSL2, and MSL3 signal enrichment in M1[WT] (♀ *msl-1−; Ubi:msl-2$^{WT}$-FLAG/ Ubi:msl-1$^{WT}$*), M1[GS] (♀ *msl-1−; Ubi:msl-2$^{WT}$-FLAG/ Ubi:msl-1$^{GS}$*), M1[Δ41–85] (♀ *msl-1−; Ubi:msl-2$^{WT}$-FLAG/ Ubi:msl-1$^{\Delta41-85}$*), M1[Δ1–15] (♀ *msl-1−; Ubi:msl-2$^{WT}$-FLAG/ Ubi:msl-1$^{\Delta1-15}$*) females.

**Figure supplement 2.** Regions with male-specific lethal (MSL)1 alone (no colocalization with MSL2) peaks (autosomes and X chromosome).

N-terminal sequences despite the presence of high homology in the dimeric coiled-coil region. Even between *Anopheles gambiae* and *Drosophila melanogaster*, the X chromosome is thought to have formed independently from the same ancestral chromosome (*Jiang et al., 2015*; *Pease and Hahn, 2012*; *Toups and Hahn, 2010*).

The generally accepted model has suggested that most MSL complexes initially bind with numerous HAS on the male X chromosome (*Kageyama et al., 2001*; *Kelley et al., 1997*; *Meller et al., 2000*;

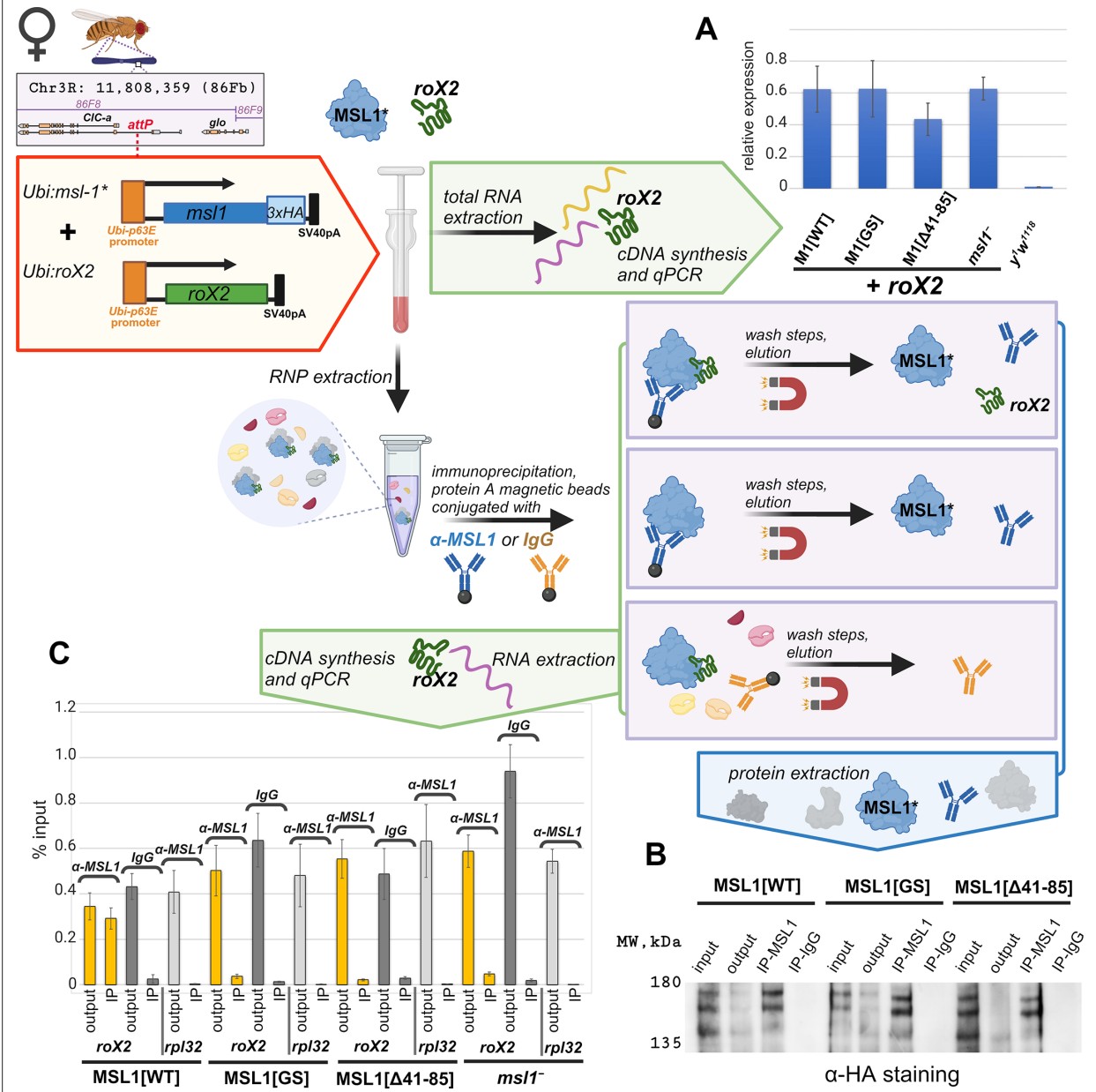

**Figure 7.** Testing role of the N-terminal region in recruiting of roX2 by male-specific lethal (MSL)1. Extraction of RNA and RNP from adult flies, and immunoprecipitation followed by RNA extraction were prepared as described in the Materials and Methods section and briefly schematically explained in the drawing. (**A**) Expression levels of the roX2 RNA in females of the $y^1w^{1118}$, M1[WT] (*msl-1⁻; Ubi:msl-1^WT-HA/ Ubi:roX2*), M1[GS] (*msl-1⁻; Ubi:msl-1^GS-HA/ Ubi:roX2*), M1[Δ41–85] (*msl-1⁻; Ubi:msl-1^Δ41-85-HA/ Ubi:roX2*) and *msl1⁻*(*msl-1⁻;Ubi:roX2/TM6,Tb*). Individual transcript levels were determined by RT-qPCR with primers for the *roX2* gene normalized relative to *RpL32* for the amount of input cDNA. The error bars show standard deviations of triplicate measurements. This panel was created using BioRender.com. (**B**) RNP extracts from adult females were immunoprecipitated with antibodies against MSL1 (IP-MSL1) or nonspecific rabbit IgG (IP-IgG) as a negative control. The efficiency of immunoprecipitation was tested by immunoblot analysis for the presence of HA-tagged MSL1 protein in immunoprecipitate samples. (**C**) Total RNA was extracted from immunoprecipitates (α-MSL1 or IgG) and analyzed for the presence of *roX2* RNA by RT-PCR. *RpL32* was used as the negative control. The results of immunoprecipitations are presented as the percentage of input cDNA. 'output' – supernatant after immunoprecipitation; 'IP' – immunoprecipitated sample. The error bars indicate SDs from three independent biological samples.

The online version of this article includes the following source data for figure 7:

**Source data 1.** Original files for the immunoblot blot analysis are in *Figure 7B*.

**Source data 2.** Files containing original immunoblots for *Figure 7B* indicate the relevant bands.

*Park et al., 2003*; *Park and Kuroda, 2001*). Most HAS contain low-complexity GAGA motifs, referred to as the MRE (*Alekseyenko et al., 2008*; *Straub et al., 2008*). The CLAMP protein binds to MREs and is responsible for chromatin opening (*Albig et al., 2019*; *Rieder et al., 2019*). MSL2 contributes to the specific recognition of HAS through the direct binding of GAGA motifs (*Villa et al., 2016*) and interactions with CLAMP (*Albig et al., 2019*; *Eggers et al., 2023*; *Tikhonova et al., 2019*). Subsequently, the MSL complexes spread from the HAS to active X chromosomal genes through distance chromatin contacts, which are formed by preexisting chromosome architecture (*Prayitno et al., 2019*; *Schauer et al., 2017*). roX RNAs guide the MSL complex assembly and its spread from HAS into flanking chromatin (*Ankush Jagtap et al., 2019*; *Ilik et al., 2017*; *Lv et al., 2019*; *Müller et al., 2020*; *Park et al., 2002*; *Valsecchi et al., 2021*; *Villa et al., 2021*). It was shown that expression of roX RNA on autosome in the presence of MSL proteins excess can induce the local spreading of MSL complex from the site of the transgene localization (*Kelley et al., 2008*; *Oh et al., 2003*; *Park et al., 2002*).

Recent studies have shown that low-complexity C-terminal domains of MSL2 and roX RNAs form a stably condensed state that induces local 'trapping' of the MSL complex (*Valsecchi et al., 2021*). Experimental evidence shows that roX2 RNA plays an instructive role by modulating the MSL complex binding specificity (*Villa et al., 2021*). For example, *Rieder et al., 2019* showed that at the initiation of embryogenesis, the MSL complex first binds to all chromosomes before becoming concentrated on the X chromosome. Weak interactions with chromatin sites on autosomes likely lead to the rapid dissociation of the MSL complex, resulting in the gradual concentration of the components of the complex on HAS. It was shown that complete MSL complexes preferentially interact with the sites on the X chromosome, while the MSL1 and MSL2 proteins can independently bind to many promoters located at all chromosomes (*Chlamydas et al., 2016*; *Hallacli et al., 2012*; *Straub et al., 2013*; *Valsecchi et al., 2018*).

The MSL1$^{GS}$ is the best for testing the functional role of the MSL1-roX interaction because MSL1$^{\Delta1-15}$ is unstable and MSL1$^{\Delta41-85}$ loses its ability to interact effectively with autosomal promoters. Thus, the N-terminal region between 41 and 85 aa is significant not only for interaction with RNA but also because plays an additional role in attracting MSL1 to active promoters independently from the MSL complex. Previous studies have shown that MSL1 alone or in combination with MSL3 or MOF also associated with many autosomal promoters independent of MSL2 (*Chlamydas et al., 2016*; *Straub et al., 2013*). Our ChIP seq data suggest that MSL1 and MSL3 are recruited to the promoters mainly independent of each other. In contrast with the recruitment of the MSL complex to HAS on the X chromosome, the binding of MSL2 to autosomal promoters does not require the MSL1 protein (*Valsecchi et al., 2018*). Thus, MSL proteins can function autonomously or as a part of alternative transcription complexes.

Our results with the MSL1$^{GS}$ mutant agree with the model that the interaction of MSL1 with roX RNAs is critical for the assembly of the MSL complex and its specific recruitment on the X chromosome. MSL1$^{GS}$ effectively binds to autosomal promoters like MSL1$^{WT}$ but disrupts the recruitment of the MSL complex on the X chromosomal HAS (*Figure 8*). As a result, males expressing MSL1$^{GS}$ have very low viability, as shown earlier for males that have both roX genes inactivated (*roX$^-$*). In *roX$^-$* males, MSL1 and MSL2 bind only to the strongest regions of the X chromosome and begin to bind to autosomal regions (*Meller and Rattner, 2002*; *Park et al., 2003*; *Stuckenholz et al., 2003*). However, overexpression of MSL1 and MSL2 can overcome a lack of roX RNAs (*Oh et al., 2003*). Consequently, the co-binding of MSL1 and MSL2 to the X chromosome is partially restored, and the viability of males is increased. We also found that the overexpression of MSL2 partially restores the viability of males expressed MSL1$^{GS}$. Probably, a high concentration of MSL2 increases the likelihood of interaction with MSL1$^{GS}$ without the help of roX RNAs.

The MSL1$^{GS}$ and MSL1$^{\Delta41-85}$ variants can efficiently heterodimerize with MSL1$^{WT}$, suggesting that efficient MSL1 dimerization does not require roX RNAs. Thus, the interaction of MSL2 with the MSL1 dimer is a critical step in the organization of the MSL complex, which is accelerated in the presence of roX RNA.

The structure of the complex has only been determined for human MSL1 and MSL2 (*Hallacli et al., 2012*), for which the interaction does not require an RNA mediator (*Wu et al., 2011*). In contrast, mutagenic analysis showed that in *Drosophila*, MSL2 preferentially interacts with the MSL1 coiled-coil domain via its RING finger (*Copps et al., 1998*). Thus, it is possible that interaction between MSL1 and MSL2 in *Drosophila* is weaker and requires roX RNAs for efficient association. Similar to the MSL1$^{GS}$

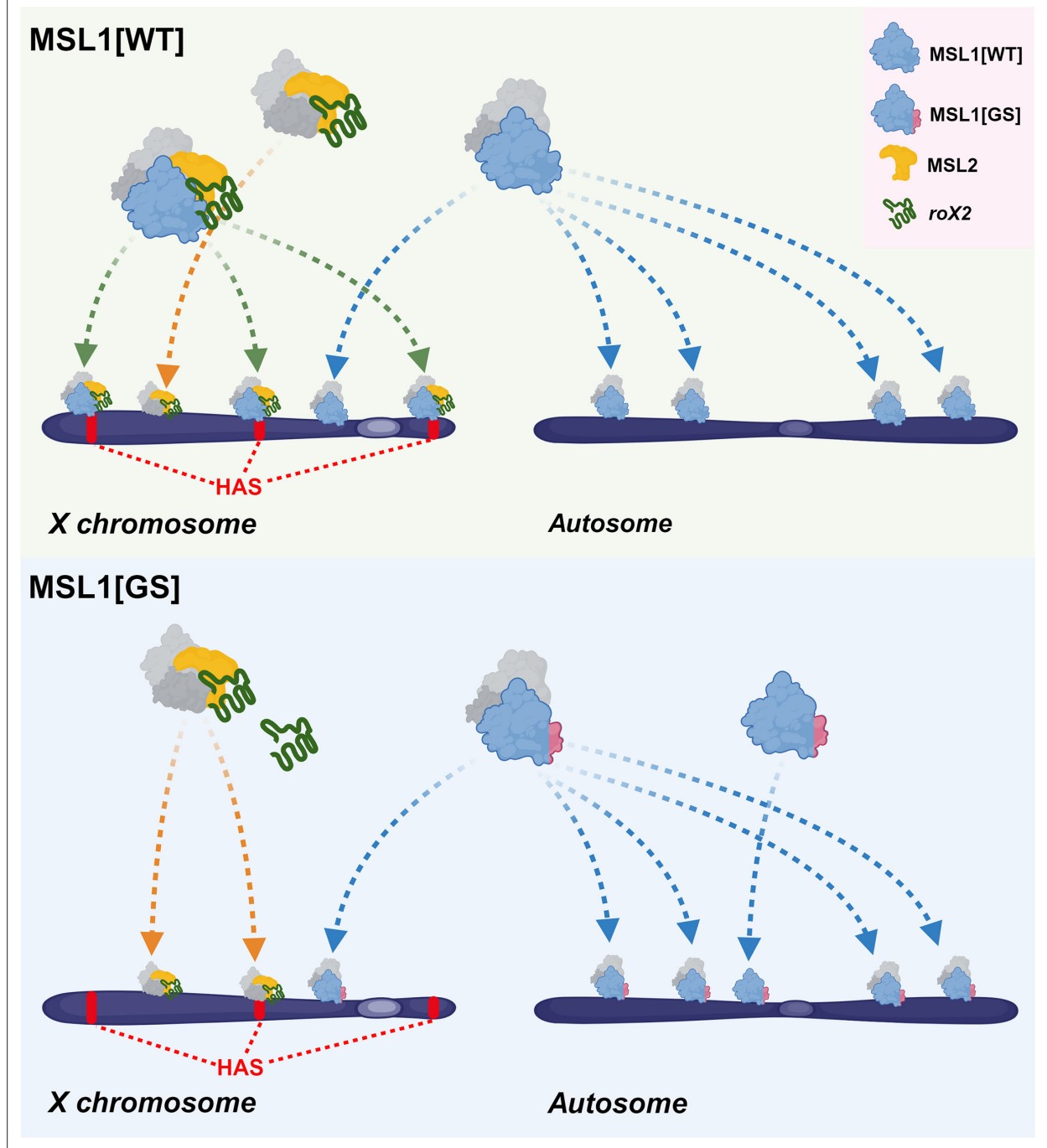

**Figure 8.** Schematic representation of male-specific lethal (MSL) complex recruiting to chromatin. The difference in the principles of recruitment of a complex containing wild-type MSL1 protein (MSL1[WT]) and a complex containing MSL1 with a mutant N-terminus is shown (MSL1[GS]). This figure was created using BioRender.com.

mutant, it has also been shown (*Hallacli et al., 2012*) that a mutant MSL1 dimer that has lost the ability to interact with MSL2 cannot effectively bind to the X chromosome.

Generally, our results are consistent with the model (*Valsecchi et al., 2021*; *Villa et al., 2021*) according to which roX RNAs provide efficient assembly of the MSL complex, which has a high affinity for HAS on the X chromosome. This process occurs through specific interactions of roX RNAs with MSL1, MSL2, MLE, and possibly MSL3 and MOF. The inability of MSL1[GS] to interact with roX RNA disrupts the formation of the MSL complex. Consequently, individual MSL proteins independently

bind to regulatory elements (mainly promoters) or integrate into alternative transcription complexes. In general, the functions of roX RNA resemble the activity of the X inactive-specific transcript (Xist) in mammals, which specifically interacts with repression complexes involved in X chromosome inactivation (**Brockdorff et al., 2020**; **Jacobson et al., 2022**). In addition, Xist specifically targets repression complexes to the inactive X chromosome. However, for *Drosophila*, the significance of roX RNAs in specific MSL recruitment to the X chromosome remains an open question requiring further examination.

# Materials and methods

## Key resources table

| Reagent type (species) or resource | Designation | Source or reference | Identifiers | Additional information |
|---|---|---|---|---|
| Genetic reagent (*Drosophila melanogaster*) | 86Fb | Bloomington *Drosophila* Stock Center | BDSC: 23648 | P{ry[+t7.2]=hsp70-FLP}1, y[1] w[*]; M{3xP3-RFP.attP}ZH-86Fb; M{RFP[3xP3.PB] GFP[E.3xP3]=vas int.B}ZH-102D |
| Genetic reagent (*D. melanogaster*) | CyO, GFP | Bloomington *Drosophila* Stock Center | BDSC: 9325 | w[1118]; sna[Sco]/CyO, P[ActGFP.w-]CC2 |
| Genetic reagent (*D. melanogaster*) | y[1] w[1118] | Bloomington *Drosophila* Stock Center | BDSC: 6598 | y[1] w[1118] |
| Genetic reagent (*D. melanogaster*) | *msl-2$^{WT}$-FLAG* | This lab (**Tikhonova et al., 2019**) | N/A | *Ubi:msl-2WT-FLAG* |
| Genetic reagent (*D. melanogaster*) | msl-1$^{L60}$ | donated by M. Kuroda | N/A | msl-1$^{L60}$ /CyO |
| Genetic reagent (*D. melanogaster*) | msl-1$^{\gamma269}$ | donated by J. Lucchesi | N/A | |
| Genetic reagent (*D. melanogaster*) | *Ubi-msl-1** | This paper | N/A | *Ubi-msl-1** transgenes are described in Materials and Methods |
| Cell line (*D. melanogaster*) | S2 | *Drosophila* genomics resource center | DGRC Stock 181; https://dgrc.bio.indiana.edu//stock/181; RRID:CVCL_Z992 | FlyBase symbol: S2-DRSC. |
| Antibody | anti-MSL1 (rabbit polyclonal) | This lab | N/A | MSL1[423–1030] IF (1:500) ChIP (1:200) WB (1:500) IP (1:200) |
| Antibody | rabbit anti-MSL2 (rabbit polyclonal) | This lab | N/A | MSL2[421-540] IF (1:500) ChIP (1:100) WB (1:500) |
| Antibody | rabbit anti-MSL3 (rabbit polyclonal) | This lab | N/A | ChIP (1:200) WB (1:500) |
| Antibody | anti-FLAG clone M2 (mouse monoclonal) | Sigma, USA | F3165 | IF (1:100) WB (1:1000) IP (1:200) |
| Antibody | anti-HA clone HA-7 (mouse monoclonal) | Sigma, USA | H9658 | IF (1:50) WB (1:500) IP (1:200) |
| Antibody | anti-lamin Dm0 clone ADL84.12 (mouse monoclonal) | DSHB, USA | ADL84.12 | WB (1:3000) |
| Antibody | anti-FLAG-HRP (mouse monoclonal) | Sigma, USA | A8592 | WB (1:1000) |

*Continued on next page*

*Continued*

| Reagent type (species) or resource | Designation | Source or reference | Identifiers | Additional information |
|---|---|---|---|---|
| Antibody | anti-mouse Alexa Fluor 555 (goat polyclonal) | Thermo Fisher Scientific, USA | A28180 | IF (1:2000) |
| Antibody | goat anti-rabbit Alexa Fluor 488 (goat polyclonal) | Thermo Fisher Scientific, USA | A11008 | IF (1:2000) |
| Chemical compound, drug | Pierce 16% Formaldehyde (w/v), Methanol-free | Thermo Fisher Scientific, USA | 28906 | |
| Chemical compound, drug | CNBr-activated Sepharose | Cytiva, UK | GE17-0430-01 | |
| Chemical compound, drug | Aminolink Plus Coupling Resin | Thermo Fisher Scientific, USA | 20505 | |
| Chemical compound, drug | TRI-reagent | MRC, USA | TR 118 | |
| Chemical compound, drug | Ribonucleoside Vanadyl Complex | NEB, USA | S1402S | |
| Chemical compound, drug | Nuclease-free BSA | Sigma, USA | 126609 | |
| Chemical compound, drug | Phase Lock Gel, QuantaBio - 2302830, Phase Lock Gel Heavy | VMR, USA | 10847–802 | |
| Chemical compound, drug | Ampure Xp beads | Beckman Coulter, USA | A63881 | |
| Chemical compound, drug | Diamant TaqA Hot-start polymerase | Belbiolab, Russia | E-TAP | |
| Chemical compound, drug | Dynabeads Protein A | Thermo Fisher Scientific, USA | 10008D | |
| Chemical compound, drug | DAPI | Applichem, Germany | A1001 | |
| Chemical compound, drug | MACSFectin | Miltenyi Biotec, USA | 130-098-410 | |
| Chemical compound, drug | SFX-Insect Cell Culture Media | HyClone, USA | SH30278 | |
| Chemical compound, drug | anti-HA magnetic beads | Thermo Fisher Scientific, USA | 88836 | |
| Chemical compound, drug | anti-FLAG magnetic beads | Sigma, USA | M8823 | |
| Chemical compound, drug | mouse IgG magnetic beads | NEB, USA | S1431 | |
| Chemical compound, drug | Halt Protease Inhibitor Cocktail | Thermo Fisher Scientific, USA | 78438 | |
| Chemical compound, drug | Calbiochem Complete Protease Inhibitor Cocktails V | Merck, USA | 539137 | |
| Chemical compound, drug | Calbiochem Complete Protease Inhibitor Cocktails VII | Merck, USA | 539138 | |
| Sequence-based reagent | Oligonucleotides used are listed in '*Supplementary file 1*' | Evrogen, Lytech, DNA synthesis | | |
| Commercial assay or kit | Crescendo Western Blotting substrate | Merck, USA | WBLUR0100 | |
| Commercial assay or kit | ChIP DNA Clean&Concentrator kit | Zymo Research, USA | D5205 | |

*Continued on next page*

*Continued*

| Reagent type (species) or resource | Designation | Source or reference | Identifiers | Additional information |
|---|---|---|---|---|
| Commercial assay or kit | NEBNext Ultra II DNA Library Prep Kit for Illumina | NEB, USA | E7645L | |
| Commercial assay or kit | Qubit dsDNA HS Assay Kit | Life Technologies Corporation | Q32851 | |
| Software, algorithm | ImageJ 1.54 f and Fiji bundle 2.14.0 | *Schindelin et al., 2012* | N/A | fiji.sc |
| Software, algorithm | cutadapt | *Martin, 2011* | N/A | |
| Software, algorithm | Bowtie version 2 | *Langmead and Salzberg, 2012* | N/A | |
| Software, algorithm | deepTools | *Ramírez et al., 2014* | N/A | |
| Software, algorithm | Picard | http://broadinstitute.github.io/picard/ | N/A | |
| Software, algorithm | MACS version 2 | *Zhang et al., 2008* | N/A | |
| Software, algorithm | R version 4.2.1 | http://www.r-project.org | N/A | |
| Software, algorithm | ChIPpeakAnno package | *Zhu et al., 2010* | N/A | |
| Software, algorithm | ChIPseeker package | *Yu et al., 2015* | N/A | |
| Software, algorithm | pheatmap package. | https://github.com/raivokolde/pheatmap; *Kolde, 2019* | N/A | |
| Software, algorithm | Gviz | *Helaers et al., 2011* | N/A | |
| Other | 100 µm MACS SmartStrainers | Miltenyi Biotec, USA | 130-110-917 | |
| Other | 70 µm MACS SmartStrainers | Miltenyi Biotec, USA | 130-110-916 | |

## Plasmid construction

PCR-directed mutagenesis was used to make constructs with deletion variants of MSL1 (corresponding primers are given in *Supplementary file 1*). Different full-sized variants of MSL1 were cloned into an expression vector containing *attB* site for φC31-mediated recombination, *Ubi-p63E* promoter with its 5'UTR, 3'UTR with SV40 polyadenylation signal, intronless *yellow* gene as a reporter for detection of transformants (*Figure 1—figure supplement 2*).

## Fly crosses and transgenic lines

*Drosophila* strains were grown at 25 °C and 75% humidity under standard culture conditions. The transgenic constructs were injected into pre blastoderm embryos using the φC31-mediated site-specific integration system at locus 86Fb (*Bischof et al., 2007*). The emerging adults were crossed with the *y ac w$^{1118}$* flies, and the progeny carrying the transgene in the 86Fb region were identified by *y$^+$* pigmented cuticle.

The *msl-1$^-$* background corresponds to *msl-1$^{γ269}$/msl-1$^{L60}$* heterozygote. The *msl-1$^{L60}$/CyO* (*Chang and Kuroda, 1998*) and *msl-1$^{γ269}$cn$^1$bw$^1$/CyO* (*Palmer et al., 1993*) stocks with the null *msl-1* mutations were kindly donated by M. Kuroda and J. Lucchesi. The *w[1118]; sna[Sco]/CyO, P[ActGFP.w$^-$] CC2* stock no. 9325 was obtained from the Bloomington stock center. To access the viability of males carrying MSL1 variants (MSL1*, MSL1$^{Δ1-15}$, MSL1$^{Δ41-85}$, and MSL1$^{Δ1-85}$), 2–3 days old virgin *msl-1$^{L60}$/CyO, GFP; Ubi-msl-1*/ Ubi-msl-1** females were crossed with 2–3 days old *msl-1$^{γ269}$/CyO, GFP; Ubi-msl-1*/Ubi-msl-1** males. *Ubi-msl-1** is any transgene expressing one of the tested MSL1 variants. *msl-1$^{L60}$/msl-1$^{γ269}$; Ubi-msl-1*/Ubi-msl-1** males and *msl-1$^{L60}$/msl-1$^{γ269}$; Ubi-msl-1*/Ubi-msl-1** females were counted after hatching. The percentage of male viability with different *Ubi-msl-1** transgenes was estimated by taking the viability of females as 100%. The viability of males carrying MSL1 variants (MSL1*, MSL1$^{WT}$, MSL1$^{Δ66-85}$ and MSL1$^{Δ24-39}$) was tested in *msl-1$^{L60}$/msl-1$^{γ269}$; Ubi-msl-1*/Ubi-msl-1** lines.

To assess the viability of females expressed MSL2 and one of MSL1 variants (MSL*), 2–3 days old virgin *Ubi-msl-1*/Ubi-msl-1** females were crossed with 2–3 days old *Ubi:msl-2^{WT}-FLAG/TM6,T* viability was estimated by taking the viability of *Ubi-msl-1*/TM6, Tb* females as 100%.

To assess the role of MSL1 variants in the binding of MSL complex to polytene chromosomes, 2–3 days old virgin *msl-1^{L60}/msl-1^{γ269}; Ubi-msl-1*/Ubi-msl-1** females were crossed with 2–3 days old *msl-1^{L60}/CyO, GFP; Ubi:msl-2^{WT}-FLAG/ TM6, Tb* males. *msl-1^{L60}/msl-1^{γ269}; Ubi-msl-1*/ Ubi:msl-2^{WT}-FLAG* larvae were selected for further study.

*b* males. *Ubi:msl-2^{WT}-FLAG* is transgene (86Fb) expressing the MSL2 cDNA under the control of the Ubi-p63E promoter at the 86F8 site (*Tikhonova et al., 2019*). The percentage of *Ubi:msl-2^{WT}-FLAG /Ubi-msl-1** female.

## Protein extract preparation

Twenty adult flies were cooled in liquid nitrogen, and homogenized for 30 s with a pestle in 200 µL of extraction buffer (20 mM HEPES, pH 7.5, 100 mM KCl, 5% glycerol, 10 mM EDTA, 1% NP-40, 1% sodium deoxycholate, 0.1% SDS, 1 mM DTT, 5 mM PMSF, and 1:100 Calbiochem Complete Protease Inhibitor Cocktails VII and V) and incubated on ice for 10 min. The suspension was sonicated in a Bioruptor (Diagenode, USA) for 3 min on setting H, 15 s ON/45 s OFF. Then 4 x SDS-PAGE sample buffer was added to the homogenate. Extracts were incubated for 10 min at 100 °C, centrifuged at 16,000 x g for 5 min, and loaded on a 6% SDS-PAGE gel.

## Immunostaining of polytene chromosomes

*Drosophila* hird instar larvae were cultured at 18 °C under standard conditions. Immunoblot analysis showed that the expression of MSL1 and MSL2 under the control of the *Ubi-p63E* promoter is significantly increased at 18 °C compared to 25 °C (*Figure 1—figure supplement 3B*). Polytene chromosome staining was performed as described (*Murawska and Brehm, 2012*). The following primary antibodies were used: rabbit anti-MSL1 at 1:500 dilution, rabbit anti-MSL2 at 1:500 dilution, mouse anti-HA at 1:50 dilution, and mouse anti-FLAG at 1:100 dilution. The secondary antibodies were Alexa Fluor 488 or 555 goat anti-rabbit and anti-mouse, at 1:2000 dilution (Invitrogen). The polytene chromosomes were co-stained with DAPI (AppliChem). Images were acquired on the Nikon Eclipse T*i* fluorescent microscope using Nikon DS-Qi2 digital camera, processed with ImageJ 1.54 f and Fiji bundle 2.14.0. 3–4 independent staining and 4–5 samples of polytene chromosomes were performed with each MSL1-expressing transgenic line. Images of stretched chromosomes are obtained as follows: based on the DAPI channel, a chromosome trace is created using the 'Segmented line' tool, the thickness is selected ('Edit-Selection-Properties'), saved as 'Selection' in.roi format. This file is opened on top of the desired channel image, and the 'Edit-Selection-Straighten' commands are applied.

## Co-immunoprecipitation assay

*Drosophila* S2 cells (obtained from DGRC, Stock 181; https://dgrc.bio.indiana.edu//stock/181) grown in SFX medium (HyClone) at 25 °C were co-transfected by plasmid expressing MSL1*–3xHA and plasmid expressing MSL2-3xFLAG with MACSFectin (Miltenyi Biotec), as recommended by the manufacturer. After transfection, the cells were incubated for 48 hr, washed once with cold 1x PBS, and resuspended in 20 packed cell volumes of hypotonic lysis buffer (20 mM Tris-HCl, pH 7.4, with 10 mM KCl, 10 mM MgCl2, 2 mM EDTA, 10% glycerol, 1% Triton X-100, 1 mM DTT, 0.5 mM PMSF and Halt Protease Inhibitor Cocktail). After incubation on ice for 10 min, the cells were sonicated in a Bioruptor (Diagenode, USA) for 2 min on setting L, 15 s ON/45 s OFF, NaCl was added to a final concentration of 420 mM, and incubation on ice continued for 60 min, with periodic mixing. Sonication was repeated as above to reduce viscosity, cell debris was pelleted by centrifugation at 10,000 g for 30 min at 4 °C, and the supernatant was collected for immunoprecipitation with anti-FLAG, anti-HA, and mouse IgG magnetic beads equilibrated in incubation buffer-150 (20 mM Tris-HCl, pH 7.4, with 150 mM NaCl, 10 mM MgCl2, 1 mM EDTA, 1 mM EGTA, 10% glycerol, and 0.1% NP-40). The protein extract (50 µg protein) was adjusted to a volume of 200 µL with buffer-150, mixed with magnetic beads (15 µL), and incubated on a rotary shaker overnight at 4 °C. The beads were then washed with five portions of buffer-150, resuspended in SDS-PAGE loading buffer, boiled, and analyzed by immunoblotting with anti-FLAG-HRP antibodies (Sigma) antibodies. Proteins were detected using the Crescendo Western Blotting substrate (Millipore).

## RNA immunoprecipitation assay

To prepare the samples, 100 mg of adult flies were ground in a mortar in liquid nitrogen and resuspended in 0.5 mL of VNC-hypotonic lysis buffer (20 mM Tris-HCl, pH 7.4, with 10 mM KCl, 10 mM MgCl2, 10% glycerol, 1% Triton X-100, 1 mM DTT, 0.5 mM PMSF, 2 mM Ribonucleoside Vanadyl Complex, and 1xHalt Protease Inhibitor Cocktail). The suspension was homogenized in a Dounce (loose pestle, 20 strokes) homogenizer and filtered through 100 µm MACS SmartStrainers (Miltenyi Biotec, United States). After incubation on ice for 10 min, the suspension was sonicated in a Bioruptor (Diagenode, USA) for 3 min on setting L (15 s ON/45 sec OFF). NaCl was added to a final concentration of 420 mM, and the suspension was incubated on ice for an additional 60 min with periodic mixing. Sonication was repeated as above to reduce viscosity. Cell debris was pelleted by centrifugation at 10,000 g for 30 min at 4 °C, and the supernatant was collected for subsequent immunoprecipitation. The RNA-protein extracts were diluted to 150 mM of NaCl with VNC-incubation buffer-0 (20 mM Tris-HCl, pH 7.4, 10 mM MgCl2, 10% glycerol, 0.1% NP-40, 0.5 mM PMSF, 2 mM Ribonucleoside Vanadyl Complex, and 1xHalt Protease Inhibitor Cocktail). Rabbit anti-MSL1 (1:200) and non-specific IgG were incubated for 1 hr at room temperature with 20 µL aliquots of Protein A Dynabeads (Thermo Fisher, USA) mixed with 200 µL of PBST. The antibody–Dynabead complexes were then washed and equilibrated in VNC-incubation buffer-150 (20 mM Tris-HCl, pH 7.4, 150 mM NaCl, 10 mM MgCl2, 10% glycerol, 0.1% NP-40, 0.5 mM PMSF, 2 mM Ribonucleoside Vanadyl Complex, and 1xHalt Protease Inhibitor Cocktail). The RNA-protein extracts (10% aliquots were stored as input material for subsequent protein and RNA extractions) were mixed with magnetic beads and incubated on a rotary shaker overnight at 4 °C. The beads were then washed with four portions of VNC-incubation buffer-150 (10% aliquots of supernatants after immunoprecipitation were stored as output material for subsequent protein and RNA extractions). Half of the beads were resuspended in SDS-PAGE loading buffer, boiled, and analyzed by immunoblotting with anti-HA-HRP antibodies (Thermo Fisher, USA). Proteins were detected using the Crescendo Western Blotting substrate (Millipore). The other half of the beads were resuspended in the TRI reagent for subsequent RNA extraction and cDNA synthesis.

## RT-PCR

Total RNA was isolated using the TRI reagent (Molecular Research Center, United States) according to the manufacturer's instructions. RNA was treated with two units of Turbo DNase I (Ambion) for 30 min at 37 °C to eliminate genomic DNA. The synthesis of cDNA was performed according to the manufacturer's instructions using 1 µg of RNA or entire sample volume after immunoprecipitation, 100 U of EpiScript reverse transcriptase (LGC Biosearch Technologies, UK), and 10 pM of random hexamers as a primer. The amounts of specific cDNA fragments were quantified by real-time PCR. At least three independent measurements were made for each RNA sample. Relative levels of mRNA expression were calculated in the linear amplification range by calibration to a standard genomic DNA curve to account for differences in primer efficiencies. Individual expression values were normalized with reference to *RpL32* mRNA.

## ChIP-seq

Chromatin was prepared from 2–3-day-old adult flies. Samples of 500 mg each of adult flies were ground in a mortar in liquid nitrogen and resuspended in 10 mL of buffer A (15 mM HEPES-KOH, pH 7.6, 60 mM KCl, 15 mM NaCl, 13 mM EDTA, 0.1 mM EGTA, 0.15 mM spermine, 0.5 mM spermidine, 0.5% NP-40, 0.5 mM DTT) supplemented with 0.5 mM PMSF and Calbiochem Complete Protease Inhibitor Cocktail V. The suspension was then homogenized in a Potter (10 strokes) and subsequently in a Dounce (tight pestle, 15 strokes) homogenizers and filtered through 70 µm MACS SmartStrainers (Miltenyi Biotec, United States). The homogenate was cross-linked with 1% formaldehyde for 15 min at room temperature. Cross-linking was stopped by adding glycine to a final concentration of 125 mM. The nuclei were washed with three 10 mL portions of wash buffer (15 mM HEPES-KOH, pH 7.6, 60 mM KCl, 15 mM NaCl, 1 mM EDTA, 0.1 mM EGTA, 0.1% NP-40, protease inhibitors) and one 5 mL portion of nuclear lysis basic buffer (15 mM HEPES, pH 7.6, 140 mM NaCl, 1 mM EDTA, 0.1 mM EGTA, 1% Triton X-100, 0.5 mM DTT, 0.1% sodium deoxycholate, and protease inhibitors), and resuspended in 1 mL of nuclear lysis buffer (15 mM HEPES, pH 7.6, 140 mM NaCl, 1 mM EDTA, 0.1 mM EGTA, 1% Triton X-100, 0.1% sodium deoxycholate, 0.5% SLS, 0.1% SDS, 0.5 mM PMSF, Calbiochem Complete Protease Inhibitor Cocktail V). The suspension was incubated for 30 min at 4 °C and then sonicated

with Covaris ME220 focused-ultrasonicator (50 alternating 15 s ON and 45 s OFF intervals, at 6 °C, peak power 75.0, duty % factor 25, cycles/burst 1000), and 50 μL aliquots were used to test the extent of sonication and to measure DNA concentration. Chromatin was then transferred to 1.5 ml-Eppendorf tubes and centrifuged at 14,000 rpm for 10 min at 4 °C. Finally, supernatants were pooled, aliquoted, and frozen in liquid nitrogen.

For immunoprecipitation chromatin was pre-cleared with Protein A Dynabeads (Thermo Fisher, USA), blocked with 0.5% BSA. Corresponding antibodies were incubated for 1 hr at room temperature with 20 μL aliquots of Protein A (rabbit anti-MSL1, 1:200; anti-MSL2, 1:100; anti-MSL3, 1:200; rabbit non-specific IgG) mixed with 200 μL of PBST. Then antibody–Dynabead complexes were washed and equilibrated in nuclear lysis buffer. Chromatin samples containing 10–20 μg of DNA equivalent in 200 μL nuclear lysis buffer (2 μL aliquots of pre-cleared chromatin as input material) were incubated overnight at 4 °C with antibody–Dynabead complexes. After three rounds of washing with lysis buffer supplemented with 500 mM NaCl and TE buffer (10 mM Tris-HCl, pH 8; 1 mM EDTA), the DNA was eluted with elution buffer (50 mM Tris-HCl, pH 8.0; 1 mM EDTA, 1% SDS), treated with RNase A for 30 min and proteinase K overnight, incubated at 65 °C for 6 hr and extracted with ChIP DNA Clean&Concentrator kit (Zymo Research, USA).

The ChIP-seq libraries were prepared with NEBNext Ultra II DNA Library Prep kit, as described in the manufacturer's instructions. In short, eluted DNA was end-repaired, and terminal adenosine residues were added using the NEBNext reagents. Indexed adapters were ligated, after which the material was size selected at ~200–600 bp with Ampure XP beads (Beckman Coulter). PCR amplification was performed using NEB primers for 15–16 cycles using the Q5 Hot Start HiFi PCR Master Mix (NEB). The PCR-amplified library was purified using Ampure XP beads and its quality was assessed on a Bioanalyzer 2100 system (Agilent). Diluted libraries were clustered on a pair-read flowcell and sequenced using a NovaSeq 6000 system (Illumina) (using «Geneti c o» facility).

## ChIP-seq data processing and sequence analysis

All ChIP-seq raw data were presented as two biological replicates with paired-end reads (except MSL1 M1[wt]♂), for which only one biological replicate was obtained. Trimming and mapping were performed using cutadapt software (*Martin, 2011*) and Bowtie version 2 (*Langmead and Salzberg, 2012*), as described previously (*Sabirov et al., 2021*). The dm6 version of the *Drosophila melanogaster* genome was used as a reference genome. After merging replicates, coverage tracks (BedGraph) were obtained using the deepTools (*Ramírez et al., 2014*) bamCoverage function, with bin widths of 10 bp and extendReads option and normalized by reads per kilobase of the transcript, per million mapped reads (RPKM). Raw and processed data were deposited in the NCBI Gene Expression Omnibus (GEO) under accession number GSE243396.

Peak calling was performed for MSL proteins from flyes of ♂M1[wt] and ♀ M1[wt]+MSL2 lines, as described previously (*Sabirov et al., 2021*). Briefly:

1. duplicates were removed with Picard (http://broadinstitute.github.io/picard/);
2. blacklist filtration was performed;
3. peak calling was performed using MACS version 2 against nonspecific IgG (*Zhang et al., 2008*), with a soft p-value threshold of $1 \times 10^{-2}$ for further irreproducible discovery rate (IDR) analysis.

Reproducibility was assessed using the IDR pipeline, with p-value thresholds of 0.05 for true replicates and 0.01 for pseudoreplicates. All samples showed ideal reproducibility both the rescue ratio (RR) and self-consistency ratio (SR) were less than 2; thus, an 'optimal' set of highly-reproduced peaks was chosen for each sample.

Further analysis was performed in R version 4.2.1 (http://www.r-project.org). Colocalization analysis was performed using the ChIPpeakAnno package (*Zhu et al., 2010*). Average signal calculation and heatmaps were constructed using the ChIPseeker package (*Yu et al., 2015*).

HAS regions for which ChIP-seq signal enrichment was calculated were obtained by combining HAS variants from three different studies (*Alekseyenko et al., 2008*; *Ramírez et al., 2015*; *Straub et al., 2008*). Only HAS regions located on the X chromosome were considered in this analysis, resulting in 301 HAS regions (*Figure 5—source data 1*). For *Figure 5—figure supplements 1A and 2A* visualization 10% autosomal regions from ♂M1[wt] with the highest log p-value were chosen for every MSL protein (MSL1 - 671, MSL2 - 288, MSL3 - 185 peaks).

To investigate MSL binding for female fly lines at different genomic locations, a list of unique MSL1, MSL2 peaks for ♀ M1[wt]+MSL2 fly line was generated using findOverlapsOfPeaks function (maxgap = 500). This list was subdivided into two groups, according to the mean logFC between protein signal and control at 500 bp region around the center of each peak: 'MSL1 alone' (MSL1 >threshold, MSL2 <threshold, threshold = 0.2) and 'MSL1 and MSL2 overlapping' (MSL1 >threshold, MSL2 >threshold, threshold = 0.4). For 'MSL1 alone' only peaks not overlapping with HAS were chosen (AUT +X) and for 'MSL1 MSL2 overlapping' X-associated peaks were selected (HAS +X). Regions were hierarchically clustered using hclust (method = 'ward.D2') and subdivided into a chosen number of clusters (*Figure 6—source data 1*) using the cutree function from package stats. Cluster visualization was made with the pheatmap package (https://github.com/raivokolde/pheatmap, copy archived at *Kolde, 2019*). Gviz (*Helaers et al., 2011*) was used for genomic track visualization.

## Acknowledgements

We thank Mitzi Kuroda (Harvard Medical School), and Erica Larschan (Brown University) for the kindly providing fly strains. We thank Farhod Hasanov and Aleksander Parshikov for fly injection. This study was performed using the equipment of the Center for Precision Genome Editing and Genetic Technologies for Biomedicine of IGB RAS supported by the Ministry of Science and Education of the Russian Federation. Some cartoons (*Figures 7 and 8*) were created with BioRender.com. This work was supported by the grant 21-14-00211 from Russian Science Foundation. Chip-Seq analysis was supported by grant 075-15-2019-1661 from the Ministry of Science and Higher Education of the Russian Federation.

## Additional information

### Funding

| Funder | Grant reference number | Author |
|---|---|---|
| Russian Science Foundation | 21-14-00211 | Oksana Maksimenko |
| Ministry of Science and Higher Education of the Russian Federation | 075-15-2019-1661 | Oksana Maksimenko |

The funders had no role in study design, data collection and interpretation, or the decision to submit the work for publication.

### Author contributions

Valentin Babosha, Evgeniya Tikhonova, Data curation, Formal analysis, Validation, Investigation, Visualization; Natalia Klimenko, Anastasia Revel-Muroz, Data curation, Software, Formal analysis, Validation; Pavel Georgiev, Conceptualization, Supervision, Writing – original draft, Writing – review and editing; Oksana Maksimenko, Conceptualization, Data curation, Formal analysis, Supervision, Funding acquisition, Validation, Investigation, Visualization, Methodology, Writing – original draft, Writing – review and editing

### Author ORCIDs

Oksana Maksimenko ⬤ https://orcid.org/0000-0003-3502-0303

Reviewer #2 (Public Review): https://doi.org/10.7554/eLife.93241.3.sa1
Author response https://doi.org/10.7554/eLife.93241.3.sa2

## Additional files

### Supplementary files

• Supplementary file 1. The list of oligonucleotides.

• MDAR checklist

## Data availability

Sequencing data have been deposited in GEO under accession code GSE243396.

The following dataset was generated:

| Author(s) | Year | Dataset title | Dataset URL | Database and Identifier |
|---|---|---|---|---|
| Maksimenko O, Revel-Muroz A | 2024 | N-terminus of *Drosophila* MSL1 is critical for dosage | https://www.ncbi.nlm.nih.gov/geo/query/acc.cgi?acc=GSE243396 | NCBI Gene Expression Omnibus, GSE243396 |

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
