## [Editor Report · eLife assessment]

This is a potentially **valuable** contribution, reporting a deletion analysis of the MSL1 gene to assess how different parts of the protein product interact with the MSL2 protein and roX RNA to affect the association of the MSL complex with the male X chromosome of *Drosophila*. However, the framework that the MSL complex mediates dosage compensation is outdated and has flaws, and the evidence is currently considered **inadequate** to support the claims. Because there are many ways to alter viability, sex-specific viability is insufficient to make claims regarding dosage compensation.

---

## [Referee Report · Reviewer #2 (Public Review)]

Summary:

A deletion analysis of the MSL1 gene to assess how different parts of the protein product interact with the MSL2 protein and roX RNA to affect the association of the MSL complex with the male X chromosome of *Drosophila* was performed.

Strengths:

The deletion analysis of the MSL1 protein and the tests of interaction with MSL2 are adequate.

Weaknesses:

This reviewer does not adhere to the basic premise of the authors that the MSL complex is the primary mediator of dosage compensation of the X chromosome of *Drosophila*. Several lines of evidence from various laboratories indicate that it is involved in sequestering the MOF histone acetyltransferase to the X chromosome but there is a constraint on its action there. When the MSL complex is disrupted, there is no overall loss of compensation but there is an increase in autosomal expression. Sun et al (2013, PNAS 110: E808-817) showed that ectopic expression of MSL2 does not increase expression of the X and indeed inhibits the effect of acetylation of H4Lys16 on gene expression. Aleman et al (2021, Cell Reports 35: 109236) showed that dosage compensation of the X chromosome can be robust in the absence of the MSL complex. Together, these results indicate that the MSL complex is not the primary mediator of X chromosome dosage compensation. The authors state that an inverse dosage effect results from a titration of the histone acetylase MOF between the NSL and MSL complexes. This is a misunderstanding of the inverse effect, which is an imbalance of regulatory molecules as described in the citation below. The inverse effect operates in triple X metafemales to produce dosage compensation of the three X chromosomes and a reduced expression of the autosomes (Sun et al 2913 PNAS 110: 7383-7388). There is no MSL complex in metafemales.

A detailed explanation was provided by Birchler and Veitia (2021, One Hundred Years of Gene Balance: How stoichiometric issues affect gene expression, genome evolution, and quantitative traits. Cytogenetics and Genome Research 161: 529-550). The relevant portions of that article that pertain to *Drosophila* are quoted below. The cited references can be found in that publication.

"In *Drosophila*, the sex chromosomes consist of an X and a Y. The Y in this species contains only a few genes required for male fertility (Zhang et al., 2020). The X consists of approximately 20% of the genome. Thus, females have two X chromosomes and males have one. Muller (1932) found that the expression of genes between the two sexes was similar but when individual genes on the X were varied in dosage they exhibited a proportional dosage effect. Each copy in a male was expressed at about twice the level as each copy in a female. Females with three X chromosomes are highly inviable but when they do survive to the adult stage, Stern (1960) found that they too exhibited dosage compensation in that the expression in the triple X genotype was similar to normal females and males. Studies in triploid flies found that dosage compensation also occurred among X; AAA, XX;AAA, and XXX; AAA genotypes via upregulation of the Xs, where X indicates the dosage of the X and A indicates the triploid nature of the autosomes (see Birchler, 2016 for further discussion). Diploid and triploid females have a similar per gene expression but the other five genotypes each must modulate gene expression by different amounts equivalent to an inverse relationship between the X versus autosomal dosage to achieve a balanced expression between the X and the A (Birchler, 1996).

Some years ago, mutations were sought in *Drosophila* that were lethal to males but viable in females. A number of such mutations were found and termed Male Specific Lethal (MSL) loci (Belote and Lucchesi, 1980). Once the products of these genes were identified, they were found to be at high concentrations on the male X chromosome (Kuroda et al., 1991). One of these genes encodes a histone acetyl transferase that acetylates Lysine16 of Histone H4 (Bone et al., 1994; Hilfiker et al., 1997). The recognition of the MSL complex and its association with the male X was an important set of contributions to an understanding of sex chromosome evolution in Drosophila (Kuroda et al., 2016). Thus, the hypothesis arose that the MSL complex accumulated this chromatin modifier on the male X to activate the expression about two-fold to bring about dosage compensation. Other data that contributed to this hypothesis were that when autoradiography of nascent transcription on salivary gland polytene chromosomes was examined in the MSL maleless mutation, the ratio of the number of grains over the X versus an autosomal region was reduced compared to the normal ratio (Belote and Lucchesi, 1980).

It has been pointed out (Hiebert and Birchler, 1994; Bhadra et al., 1999; Pal Bhadra et al., 2005; Sun et al., 2013a; Birchler, 2016), however, that the grain counts over the X and the autosomes when considered in absolute terms rather than as a ratio show that the X more or less retained dosage compensation and the autosomal numbers are about doubled, i.e. exhibit an inverse dosage effect. The same situation occurs with the msl3 mutation (Okuno et al., 1984), another MSL gene, in that the autoradiographic grain numbers as an absolute measure show retention of X dosage compensation and an autosomal increase. The data treatment to produce an X to A ratio seemed reasonable in the context of the time when all regulation in eukaryotes was considered positive. However, when studies were conducted in such a manner as to assay the absolute effect on gene expression in the maleless mutation, in adults (Hiebert and Birchler, 1994), larvae (Hiebert and Birchler, 1994; Bhadra et al., 1999; 2000; Pal Bhadra et al., 2005), and embryos (Pal Bhadra et al., 2005), the trend was for retention of dosage compensation of X linked genes and an increase in expression of autosomal genes.

In global studies, if the X to autosomal expression does not change between mutant and normal, one can conclude that dosage compensation is operating. However, a lower X to A ratio could be a loss of compensation or an increased transcriptome size from the increase of the autosomes, as suggested by the absolute data of Belote and Lucchesi (1980) and Okuno et al (1984) and that was visualized directly in embryos (Pal Bhadra et al., 2005). The transcriptome size in aneuploids can change, which cannot be detected in RNA-seq analyses alone (Yang et al., 2021), so it is an important consideration for studies of dosage compensation. It was recently acknowledged that in MSL2 knockdowns the relative X expression is decreased and a moderate autosomal increase is found (Valsecchi et al., 2021b). A similar trend is evident in the microarray data on MSL2 knockdown in SL2 tissue culture cells (Hamada et al., 2005) and in the roX RNA (noncoding RNAs essential for MSL localization on the male X) mutants (Deng and Meller, 2006). This trend is in fact consistent with the absolute data that suggest an increase in the transcriptome size (Figure 7). A global change in transcriptome size can cause a generalized dosage compensation of a single chromosome to appear as a proportional dosage effect (loss of compensation) to some degree (Figure 7).

Examination of expression in triple X metafemales, where there is no MSL complex, found that X-linked genes generally show dosage compensation but there is a generalized inverse effect on the autosomes, which could account for the detrimental effects of metafemales (Birchler et al., 1989; Sun et al., 2013b). An examination in metafemales of alleles of the white eye color gene that do or do not exhibit dosage compensation in males, showed the same response, namely, increased expression if there was no dosage compensation in males and no difference from normal females for the male dosage-compensated alleles (Birchler, 1992). This experiment demonstrated a relationship between the mechanism of dosage compensation in males and metafemales and implicated the inverse dosage effect in both. An involvement of the inverse effect in *Drosophila* dosage compensation provides an explanation for how the five levels of gene expression can be explained (Birchler, 1996), whereas an all-or-none presence of a complex on the X does not. The stoichiometric relationship of regulatory gene products provides a means to read the relative dosage at multiple doses to produce the appropriate inverse level.

What then is the function of the MSL complex? It was discovered that the MSL complex will actually constrain the effect of H4 lysine16 acetylation to prevent it from causing an overexpression of genes (Bhadra et al., 1999; 2000; Pal Bhadra et al., 2005; Sun and Birchler 2009; Sun et al., 2013a). Indeed, in the chromatin remodeling Imitation Switch (ISWI) mutants, the male X chromosome was specifically overexpressed suggesting that its normal function is needed for the constraint to occur (Pal Bhadra et al., 2005). Independently, the Mtor nuclear pore component shows a similar specific male X upregulation when Mtor is knocked down and this effect was shown to operate on the transcriptional level (Aleman et al., 2021). Interestingly, the increased expression of the X in the Mtor knockdown is accompanied by an inverse modulation of a substantial subset of autosomal genes, illustrating why the constraining process evolved to counteract male X overexpression. The constraining effect might involve a number of gene products (Birchler, 2016) and is an interesting direction for further study.

Furthermore, when the H4Lys16 acetylase was individually targeted to reporter genes, there was an increase in expression (Sun et al., 2013a). However, when other members of the MSL complex were present in normal males or ectopically expressed, this increase did not occur (Sun et al., 2013a). It thus appears that the function of the MSL complex is to sequester the acetylase from the autosomes and constrain it on the X (Bhadra et al., 1999; 2000; Pal Bhadra et al., 2005; Sun and Birchler, 2009; Sun et al., 2013a). Indeed, in the Mtor knockdowns, the X linked genes with the greatest upregulation were those with the greatest association with the acetylase and the H4K16ac histone mark (Aleman et al 2021), supporting the idea of a constraining activity that becomes released in the Mtor knockdown. When the MSL complex is disrupted, there is an inverse effect on the autosomes that occurs but in normal circumstances the sequestration mutes this effect. The MSL complex disruption releases the acetylase to be uniformly distributed across all chromosomes as determined cytologically (Bhadra et al., 1999) or via ChIPseq for H4Lys16ac (Valsecchi et al., 2021a). Indeed, the quantity of the H4Lys16ac mark only has a proportional effect on gene expression when the constraining activity is disrupted (Aleman et al., 2021) or when the MSL complex is not present (Sun et al., 2013a). Thus, in normal flies there is a more or less equalized expression of the X and autosomes despite the monosomy for 20% of the genome.

The component of the complex that is expressed in males and thought to organize the complex to the male X, MSL2, was recently found to also be associated with autosomal dosage sensitive regulatory genes (Valsecchi et al., 2018). MSL2 was found to modulate these autosomal dosage sensitive genes in various directions, which illustrates that MSL2 has a role in dosage balance that goes beyond the X chromosome. This finding is consistent with the evolutionary scenario that the initial attraction of the complex to the X chromosome was to upregulate dosage sensitive genes in hemizygous regions as the progenitor Y became deleted for them, with the constraining activity evolving to prevent an overexpression as the amount of acetylase on the male X increased with time (Birchler, 2016).

The MSL hypothesis takes an X-centric view that does not accommodate what is now known about dosage effects across the whole genome. The idea that dissolution of the MSL complex would cause reduction in expression of the male X linked genes without any consequences for the autosomes is not consistent with current knowledge of gene regulatory networks and their dosage sensitivity. Indeed, the finding of dosage compensation in large autosomal aneuploids that operates on the transcriptional level (Devlin et al., 1982; 1984; Birchler et al., 1990; Sun et al., 2013c) as well as a predominant inverse effect by the same (Devlin, et al., 1988; Birchler et al., 1990) argues that one must consider the inverse effect for an understanding of the evolution of dosage compensation in *Drosophila* (and other species). Further discussion of models of *Drosophila* compensation has been published (Birchler, 2016).

What is likely to be the most critical issue with sex chromosome evolution is the consequences for dosage sensitive regulatory genes. This fact is nicely illustrated by the retention of these types of genes in different independent vertebrate sex chromosome evolutions (Bellott and Page, 2021). In *Drosophila*, by contrast, dosage compensation is more of a blanket effect on most but not all X linked genes despite the fact that many genes on the X are unlikely to have dosage detrimental effects, although dosage sensitive genes might have played a role as noted above. The particularly large size of the X in *Drosophila* compared to the whole genome is potentially a contributing factor because such large genomic imbalance is likely to modulate most genes across the genome. Also, there is no evidence of a WGD in *Drosophila* as there is in other species for which the inverse effect has been documented (maize, Arabidopsis, yeast, mice, human). These other species have various numbers of retained duplicate dosage sensitive regulatory genes from WGDs. Thus, the relative change of regulatory genes in aneuploids in these species will not be as great compared to some of their interactors in the remainder of the genome, which could result in lesser magnitudes of some trans-acting effects, similarly to how aneuploids in ascending ploidies have fewer effects as described above. The absence of duplicate regulatory genes in Drosophila would predict a stronger inverse effect in general and that could have been capitalized upon to produce dosage compensation of most genes on the X chromosome despite many of them not being dosage critical. While sex chromosome evolution must accommodate dosage sensitive genes for proper development and viability, it could also be capitalized upon to evolve sexual dimorphisms in expression (Sun et al., 2013c)."

Comments on revised submission:

The authors did make an effort to address the issue previously raised.

The authors state that an inverse dosage effect results from a titration of the histone acetylase MOF between the NSL and MSL complexes (lines 87-89). This is a misunderstanding of the inverse effect, which is an imbalance of regulatory molecules. Single regulatory gene dosage series can produce this effect. The inverse effect operates in triple X metafemales to produce dosage compensation of the three X chromosomes and a reduced expression of the autosomes (Sun et al 2913 PNAS 110: 7383-7388). There is no MSL complex in metafemales.

---

## [Author Response]

The following is the authors’ response to the original reviews.

Thank you for taking the time to review our manuscript. We are grateful to reviewer #1 for positive evaluation of our work and for providing valuable comments that will significantly enhance the presentation of our results. We understand reviewer #2's negative assessment because we did not discuss an alternative model of dosage compensation in *Drosophila*. We will address this omission in the Introduction section of the revised manuscript and remove any controversial statements from other parts of the text. However, it is important to clarify that our study does not focus on the mechanisms of dosage compensation. The main goal of the manuscript was to investigate the assembly of the MSL complex and its specific binding to the *Drosophila* X chromosome. We utilized male survival data to demonstrate the efficacy of MSL complex binding to the X chromosome, a relationship that has been supported by numerous independent studies. We understand that Reviewer #2 agrees that disruption of the MSL complex binding results in male lethality. As far as we understand, Reviewer #2 suggests that the MSL complex does not activate transcription of X chromosome genes, but instead facilitate the recruitment of MOF protein and potentially other general transcription factors to the X chromosome. This could explain the decrease in autosomal gene expression due to a reduction in activating factors like MOF at autosomal promoters. In the upcoming revision, we aim to strike a balance between the two models that elucidate dosage compensation in *Drosophila*. We appreciate your feedback and look forward to enhancing the clarity and coherence of our manuscript based on your insightful comments.

**Reviewer #2 (Public Review):**
Summary:A deletion analysis of the MSL1 gene to assess how different parts of the protein product interact with the MSL2 protein and roX RNA to affect the association of the MSL complex with the male X chromosome of *Drosophila* was performed.Strengths:The deletion analysis of the MSL1 protein and the tests of interaction with MSL2 are adequate.

We thank the reviewer for the positive assessment of the experimental work done.

This reviewer does not adhere to the basic premise of the authors that the MSL complex is the primary mediator of dosage compensation of the X chromosome of *Drosophila*.

We completely agree with this reviewer's claim. In the Introduction section we attempted to make clear that there are two models for the functional role of specific recruitment of the MSL complex to the X chromosome in males.

Several lines of evidence from various laboratories indicate that it is involved in sequestering the MOF histone acetyltransferase to the X chromosome but there is a constraint on its action there. When the MSL complex is disrupted, there is no overall loss of compensation but there is an increase in autosomal expression. Sun et al (2013, PNAS 110: E808-817) showed that ectopic expression of MSL2 does not increase expression of the X and indeed inhibits the effect of acetylation of H4Lys16 on gene expression. Aleman et al (2021, Cell Reports 35: 109236) showed that dosage compensation of the X chromosome can be robust in the absence of the MSL complex. Together, these results indicate that the MSL complex is not the primary mediator of X chromosome dosage compensation. The authors use sex-specific lethality as a measure of disruption of dosage compensation, but other modulations of gene expression are the likely cause of these viability effects.

Sun et al (2013, PNAS 110: E808-817) showed that recruitment of the MSL complex-specific subunit MSL2 or the MOF protein to the UAS promoter resulted in recruitment of the entire MSL complex in males but not transcriptional activation. This important result argues that the MSL complex does not activate transcription. However, it must be taken into account that the GAL4 DNA binding region used to recruit the chimeric MSL2 protein to the UAS promoter was directly fused to the MSL2 RING domain, which is critical for interaction of MSL2 with MSL1 and its ubiquitination activity (this activity could potentially be involved in transcription activation). It also remains poorly understood what happens to the MSL complex after recruitment to the promoters or HAS on the X chromosome. Subcomplex MSL1/MSL3/MOF can acetylate TF and H4K16 during RNA polymerase II elongation, resulting in increasing of transcription. The separate role of MSL2 and MSL1 in the activation of transcription of gene promoters is also shown. Sun et al. showed that in females, recruitment of MOF to the UAS promoter leads to a strong increase in transcription, which is associated with the inclusion of MOF in the non-specific lethal (NSL) complex, which is bound to promoters and is required for strong transcription activation. In males, MOF is preferentially recruited to the UAS promoter in the full MSL complex or perhaps in the MSL1/MSL3/MOF subcomplex, which stimulates transcription during RNA polymerase II elongation much less strongly than NSL complex. The same result was obtained in the Prestel et al. 2010 (Mol Cell 38:815-26). In this study the GAL4 binding sites were inserted upstream of the lacZ and mini-white genes. Activation of transcription after recruitment of GAL4-MOF to the GAL4 sites was studied in males and females. As in Sun et al. 2013, strong activation of the reporter was observed in females. A weak transcriptional activation of the reporter gene in males was shown, and the MOF protein was detected not only on the promoter, but also in the coding and 3’ regions of the reporter.

We do not understand how the paper by Aleman et al (Cell Reports 35: 109236, 2021) is consistent with the hypothesis that the MSL complex is not involved in the transcriptional activation of X chromosomal genes. The main conclusions of this paper: (1) Inactivation of Mtor leads to selective activation of the male X chromosome. (2) Mtor-driven attenuation of male X occurs in broad domains linked by the MSL complex. (3) Mtor genetically interacts with MSL components and reduces male mortality; (4) Mtor restrains dose-compensated expression at the level of nascent transcription. Thus, the paper shows that the MSL complex has an activator activity that is partially inhibited by Mtor. Accordingly, inactivation of Mtor only partially restored the survival of males in which dosage compensation was not completely inactivated.

A detailed explanation was provided by Birchler and Veitia (2021, One Hundred Years of Gene Balance: How stoichiometric issues affect gene expression, genome evolution, and quantitative traits. Cytogenetics and Genome Research 161: 529-550).

We agree that an alternative model of the dosage compensation mechanism is reasonable. We can assume that both mechanisms can function jointly provide effective dosage compensation in *Drosophila* males. At the suggestion of the reviewer to reconsider the entire context of the article, we will make many small changes throughout the manuscript.

**Reviewer #1 (Recommendations For The Authors):**
Overall, I found the text well written and the figures logically organized (especially Figure 5, which had the potential to confuse). The authors especially excelled in bringing together the decades of literature in the Discussion.I offer several suggestions to improve the readability:Consider presenting the coiled-coil domain homology in Figure 1A as a contrast for the N-terminal region, which the authors claim is poorly conserved.

We added the coiled-coil domain homology in Figure 1A in new version of the manuscript.

It is difficult to visualize the red MSL2 in Figure 2; the green and red panels should be presented separately in the main text, as they are in the Supplemental Figure 2.

We prepared Figure 2 with separate green and red panels.

The ChIP-seq experiments for MSL proteins are well presented, but in my opinion, add little to the overall conclusions:Figure 6 mostly recapitulates what has already been published and utilized by several groups, most recently the authors themselves (Tikhonova 2019): that MSL expressed in females targets the X/HAS, similar to in males. While these are nice supporting data for the female transgenic system, I do not believe this figure should be prominently featured as if this is a novelty of the current study.

We fully agree with the reviewer's comment about the limitation of scientific novelty in Figure 6. It has an auxiliary meaning. Therefore, we transferred this figure to Supplementary material (as supplement for Figure 5).

The ChIP experiments in Figure 7 agree with the conclusions in Figures 2 and 3 (polytene chromosome immunostaining) when it comes to X/autosome localization. I believe it would help with the flow of the paper if these experiments were combined or at least placed closer together in the narrative, rather than falling at the end.

We moved Figure 7 (in new version – Figure 5) closer to polytene chromosome immunostaining. We agree with reviewer that this placement of the figure will make it easier to perceive the meaning of the article as a whole.

I find Figure 8 difficult to understand, especially since the "clusters" are not annotated in the figure, but are described in the text. I struggled to follow the authors' conclusions based on these data. The authors could clarify the figure with annotations, although to be honest I do not currently see the value of this analysis/figure.

In the new version of the article, we changed this part: we removed clusters for autosomes as difficult for understanding and non-important for this manuscript. Also we tried to place emphasis more clearly in the text of the article for clusters 1 and 2 that characterize HAS.